# MATA: A Trainable Hierarchical Automaton System for Multi-Agent Visual Reasoning

**Zhixi Cai, Fucai Ke, Kevin Leo, Sukai Huang, Maria Garcia de la Banda, Peter J. Stuckey, Hamid Rezatofighi**
Monash University, Australia
`{zhixi.cai,fucai.ke1,kevin.leo,sukai.huang,maria.garciadelabanda,`
`peter.stuckey,hamid.rezatofighi}@monash.edu`

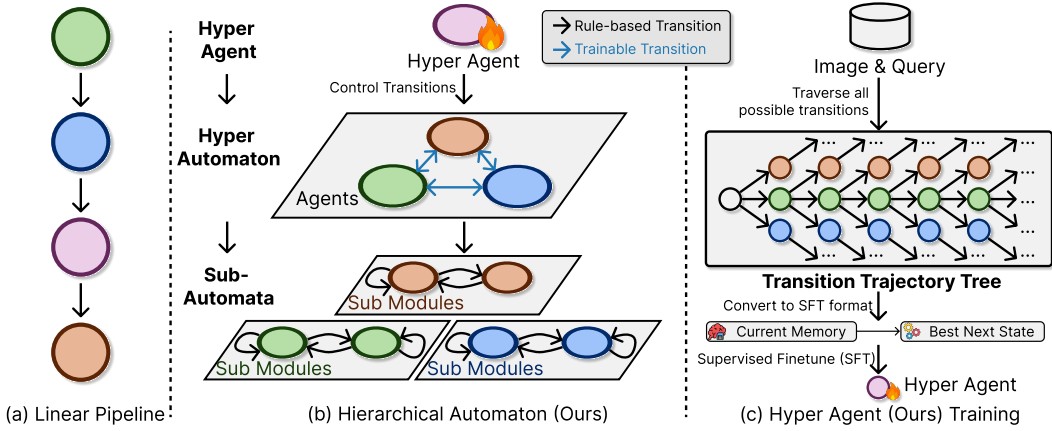

Figure 1: **Overview of MATA. (a)** Linear pipelines (previous methods) execute modules in a fixed, manually designed order. **(b)** MATA organizes agents as states in a hyper automaton. A trainable hyper agent learns high-level transitions between agents (blue arrows), enabling collaboration and competition, while each agent runs a small rule-based sub-automaton for reliable micro-control (black arrows). **(c)** To train the hyper agent, we expand a transition-trajectory tree per image-query, score the leaves using task metrics, and convert each node's snapshot into a supervised pair *current memory → best next state* for supervised finetuning (SFT), forming MATA-SFT-90K.

## ABSTRACT

Recent vision-language models have strong perceptual ability but their implicit reasoning is hard to explain and easily generates hallucinations on complex queries. Compositional methods improve interpretability, but most rely on a single agent or hand-crafted pipeline and cannot decide when to collaborate across complementary agents or compete among overlapping ones. We introduce MATA (Multi-Agent hierarchical Trainable Automaton), a multi-agent system presented as a hierarchical finite-state automaton for visual reasoning whose top-level transitions are chosen by a trainable hyper agent. Each agent corresponds to a state in the hyper automaton, and runs a small rule-based sub-automaton for reliable micro-control. All agents read and write a shared memory, yielding transparent execution history. To supervise the hyper agent's transition policy, we build transition-trajectory trees and transform to memory-to-next-state pairs, forming the MATA-SFT-90K dataset for supervised finetuning (SFT). The finetuned LLM as the transition policy understands the query and the capacity of agents, and it can efficiently choose the optimal agent to solve the task. Across multiple visual reasoning benchmarks, MATA achieves the state-of-the-art results compared with monolithic and compositional baselines. The code and dataset are available at `https://github.com/ControlNet/MATA`.

# 1 INTRODUCTION

Visual reasoning is the cognitive process of interpreting and analyzing relationships among entities in a visual scene to support decision-making and problem-solving (Ke et al., 2025b). Although recent Vision-Language Models (VLMs) (Liu et al., 2023a; Chen et al., 2024; Bai et al., 2025b) demonstrate strong perceptual ability, their implicit reasoning is difficult to audit and often causes hallucinations on complex queries involving spatial relations, spatial attributes, and counting. Compositional approaches (Surís et al., 2023; You et al., 2023; Ke et al., 2024; Cai et al., 2025) improve interpretability by decomposing a task into planning, perception, and reasoning stages, typically employing Large Language Models (LLMs) (Gemini-Team, 2023; OpenAI, 2024; DeepSeek-AI, 2025) as planners or code generators and Vision Foundation Models (VFMs) (Radford et al., 2021; Liu et al., 2023b; Xiao et al., 2024; Yang et al., 2024) as perceptual tools. Despite these improvements, non-agentic compositional methods (Surís et al., 2023; Lu et al., 2023) struggle in practice: they are limited to a single-turn reasoning, thus lacking the ability to incrementally reason in a closed-loop. Due to these limitations, agentic methods (You et al., 2023; Ke et al., 2024; Gao et al., 2024; Zhong et al., 2025) treat visual reasoning as a multi-step feedback loop in which agents actively take actions based on the current state (Ke et al., 2025b).

However, most agentic systems still employ a single agent, which is often insufficient for complex reasoning (Wang et al., 2025c) as different skills are required for different parts of a problem. Further, in prior multi-agent methods (Hong et al., 2023; Li et al., 2024; Nguyen et al., 2025; Zhang et al., 2025) (developed for other domains), *collaborative* agents are assigned disjoint roles for different subtasks and are organized into hard-coded pipelines. While this is simple and interpretable, it prevents error and hallucination handling, and tends to propagate upstream mistakes through the pipeline (Gao et al., 2024; Ke et al., 2025a). In contrast, a *competition* mechanism where functionally overlapping agents for the same subtask work together is under-explored in previous work. In this paper, we explore compositional multi-agent visual reasoning in an environment where collaborative and competitive agents exist.

Motivated by the requirements above, we cast this decision problem as a finite-state automaton where the transition function picks a discrete next state and the lifecycle is naturally expressed by explicit states and transitions. This provides explainability, verifiable control flow, and modularity that yield greater versatility, reliability, and performance. A recent work (Cai et al., 2025) also used an automaton, but its hand-written rule-based transitions are inflexible and difficult to manually define as states and transitions grow (Wang et al., 2025a; Yue et al., 2025; Dang et al., 2025; Wan et al., 2025). When new agents are added, their transitions need to be manually defined. Designing rules to select among functionally overlapping (competitive) agents is hard since the criteria are ambiguous and task-dependent, and human priors about which agents fit which tasks and queries are uncertain. We therefore design a trainable hyper agent to learn a transition policy that selects the next state. Notably, not every transition needs learning: within an agent, micro-steps (e.g., LLM/VLM prompting, verifier checks, tool I/O) follow clear procedures that are easy to define. As the number of agents grows, the main difficulty is cross-agent transition rather than agent's inside control. This motivates a hierarchical automaton in which each top-level state is an agent with a small rule-based sub-automaton, and a trainable hyper agent provides the transition function that observes the shared memory and selects the next agent. All agents read and write to a shared memory that records variables, tool outputs, code history, and verifier feedback, recording an explainable process. This replaces an inflexible rule-based transition policy with a data-driven, error-aware, and dynamic policy that can redirect to alternative solutions when needed. This design focuses on learning the ambiguous selection between competitive agents, while preserving reliable execution inside agents.

We introduce these ideas in MATA (Multi-Agent hierarchical Trainable Automaton), a hierarchical automaton for visual reasoning. MATA contains a specialized agent for fast, System 1-style perception (e.g., object detection, simple question answers); a slow, System 2-style step-wise reasoner that generates and executes Python programs for multi-step inference; and a one-shot workflow reasoner that solves queries without iteration.

To supervise the hyper agent, we need labeled transition decisions. We therefore run the system for each image-query pair, expand a transition trajectory tree (Kearns et al., 2002) and log the state history, prompts, intermediate artifacts (detections, captions, code), feedback, and performance results.

The leaves are scored by the appropriate task performance, and each decision is labeled with the child that leads to the highest-scoring subtree. This generates memory-to-next-state pairs (MATA-SFT-90K) for LLM supervised finetuning (SFT), as shown in Figure 1 (c).

The contributions of our paper are:

- A hierarchical deterministic finite-state automaton-based system, MATA, that unifies neuro-symbolic framework with collaborative and competitive multi-agent design for visual reasoning.

- Proposing (i) a learnable mechanism that trains a hyper agent as the transition policy of the hyper automaton over collaborative and competitive agents; (ii) a transition-trajectory data generation pipeline and the dataset, MATA-SFT-90K, for supervised finetuning (SFT) of the hyper agent.

- Comprehensive experiments across visual-reasoning benchmarks, with extensive ablations and analysis.

## 2 RELATED WORKS

Monolithic vision-language models (VLM) map images and text directly to answers with a single forward pass (Xiao et al., 2024; Liu et al., 2023b; Li et al., 2023a;b; Wu et al., 2023; Stanić et al., 2024; Zhu et al., 2023). While these models have strong perceptual capabilities, their implicit reasoning processes are hard to explain and often degrade on queries requiring spatial relations, counting, or multi-step reasoning (Jahangard et al., 2024; 2025). This motivates modular designs that expose intermediate, explainable symbolic processes (Andreas et al., 2016; Hsu et al., 2023). Compositional methods decompose a task into multiple stages (Ke et al., 2025b), often by having an LLM generate grounded actions (e.g., programs or JSON) executed by tools (Gupta & Kembhavi, 2023; Surís et al., 2023; Shen et al., 2023; Lu et al., 2023). These pipelines improve interpretability and enable external tools use, but usually operate in a single forward pass with a fixed manually designed pipeline. They thus lack a flexible mechanism to engage in multi-step reasoning from feedback.

Recent works (You et al., 2023; Ke et al., 2024; Gao et al., 2024; Zhong et al., 2025) explore agentic systems where an LLM/VLM reasons in multiple steps and calls tools (Ke et al., 2025b). However, most agentic approaches in visual reasoning remain single-agent. In broader domains, multi-agent frameworks assign disjoint roles and connect them with hand-crafted collaboration patterns (Hong et al., 2023; Li et al., 2024; Nguyen et al., 2025; Zhang et al., 2025), achieving better performance in reasoning. However, this idea is still under-explored for visual reasoning. Notably, noise from perception and LLM/VLM hallucinations can accumulate across steps (Ke et al., 2025a) from the collaborating pipelines, and most designs overlook competition between functionally overlapping agents (Wang et al., 2025c). This lack of a learned transition policy limits flexibility and robustness on complex and diverse queries.

Finite-state automata as abstractions provide explicit control flow and interpretability. NAVER introduces probabilistic logic inside an automaton and equips modules with self-correction (Cai et al., 2025), but relies on a hand-crafted transition policy that is hard to manually define as states grow. HYDRA introduces an agent that includes a planner, an RL controller, and a code-executing reasoner (Ke et al., 2024). While data-driven, it still focuses on instruction-level planning without a learned, high-level policy for switching across qualitatively different agents on demand. By contrast, we propose MATA that explicitly learns the inter-agent transition function over a hyper-automaton whose states are agents, while keeping intra-agent micro-steps rule-based. This learned transition function enables collaboration and competition among overlapping experts and transfers across different domains and tasks (section 4.2), which previous visual reasoning methods with hand-written transitions or single-agent controllers do not address. States are agents; each agent runs a small, rule-based sub-automaton for reliable micro-control, while a trainable hyper agent learns cross-agent transitions over a shared memory. This hierarchical view retains the interpretability of explicit state machines, avoids hand-coded transitions, and supports both collaboration and competition. Unlike prior work (Ke et al., 2024; Cai et al., 2025), our controller is supervised-trained from transition-trajectory data to transit between agents and to report a final result only when it is certain of the answer, directly addressing the gap identified above.

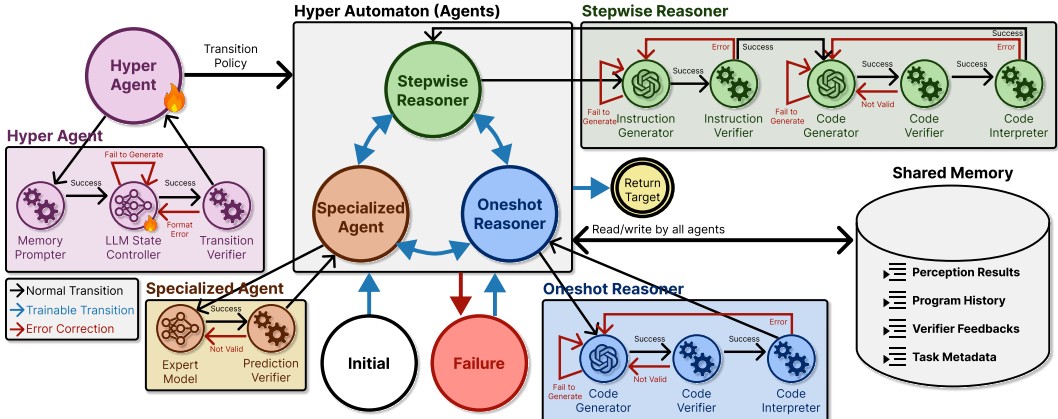

Figure 2: **Pipeline of MATA.** A trainable *hyper agent* reads a snapshot of the shared memory, predicts the next state with an *LLM State Controller*. Its decision (blue arrows) routes control among agent states in the *hyper automaton*: Oneshot Reasoner, Stepwise Reasoner, and Specialized Agent. Each agent runs a rule-based sub-automaton that iterates until return to the hyper automaton. All agents read/write an append-only *Shared Memory*, enabling the hyper agent to access the current context for choosing the optimal next state. Lifecycle states INITIAL and FAILURE are shown outside the agents (see subsection 3.2 for details).

## 3 METHODOLOGY

We explore multi-agent visual reasoning by learning a high-level transition function over agents within a hierarchical automaton, enabling data-driven *collaboration* and *competition* among overlapping skills and replacing inflexible hand-written pipelines.

### 3.1 OVERVIEW

A visual reasoning instance is an image-query pair $(v, q)$ mapped to an output $y$ (Ke et al., 2025b). MATA organizes inference as a *hierarchical automaton* operated by a trainable *hyper-agent*. Informally, the hyper automaton $\mathcal{M}_\theta$ is a top-level automaton whose states include a set of sub-agents, with each sub-agent running a small rule-based sub-automaton, and the trainable hyper agent controlling the learned transition $\delta_\theta$. Formally, it can be described as a Mealy machine (Mealy, 1955): $\mathcal{M}_\theta = (S, S_0, \Sigma, \Lambda, \delta_\theta, \Gamma)$ where $S$ denotes the set of states (containing both agent states for task execution and lifecycle states for process coordination), $S_0$ the initial state where reasoning begins, $\Sigma$ the inputs drawn from shared-memory snapshots (storing intermediate results from agents), $\Lambda$ the answer space of visual reasoning queries (e.g., discrete labels, bounding box coordinates, or free-text responses), $\delta_\theta$ the learned transition function that determines the next state based on the current state and memory inputs, and $\Gamma$ the output function that generates the final answer $\hat{y}$ once the automaton reaches a terminal state. Detailed breakdowns of the states, transition mechanics, and output generation process are provided in the subsequent sections (Figure 2).

### 3.2 HYPER AUTOMATON

**States.** The finite state set is the union of *agent states* and *lifecycle states*: $S = S_{\text{agent}} \cup S_{\text{life}}$, where $S_{\text{agent}} = \{\text{ONESHOT}, \text{STEPWISE}, \text{SPECIALIZED}\}$, $S_{\text{life}} = \{\text{INITIAL}, \text{FINAL}, \text{FAILURE}\}$ and the initial state $S_0 = \text{INITIAL}$. Agent states invoke concrete skills; lifecycle states orchestrate the progression and termination of the reasoning episode (e.g., starting the task, handling uncertainty, concluding with an answer). Details of the states are shown in Table 1.

Agents in our system are intentionally both *collaborative* and *competitive*. When control transitions from one agent to another, the successor agent reads the shared memory containing the prior history and feedback, and builds on that context; this is *collaboration*. At the same time, multiple agents may attempt the same task; if one agent stalls or fails, another can take over and complete it; this is *competition*. The learned transition policy $\delta_\theta$ selects among them based on context (e.g., ONESHOT

Table 1: **States of the hyper automaton.** The table specifies the description and the triggering condition for each state. $\delta_\theta$: transition function of hyper automaton.

| State | Description | Selected by |
|---|---|---|
| INITIAL | The unique state where reasoning begins. | Initial state |
| ONESHOT | A workflow agent that executes a single-pass program generation and execution workflow for solvable queries, equipped with a lightweight verifier. | |
| STEPWISE | A stepwise reasoner that produces step-wise Python programs for complex queries; code is verified and executed in a sandboxed environment to ensure correctness. | $\delta_\theta$ |
| SPECIALIZED | An expert agent that performs fast perception tasks; built-in verifiers validate outputs and adapt parameters. | |
| FINAL | A terminal state in which sufficient evidence has accumulated; the output function $\Gamma$ is invoked when in this state to produce the final answer $\hat{y}$. | |
| FAILURE | A state triggered by unrecoverable errors or exceeding iteration limit. | Error occurs |

vs. STEPWISE for moderately compositional VQA; SPECIALIZED vs. ONESHOT for grounding with simple perception). This overlap is intentional, as the three agents represent a spectrum: *perception (system 1)*, *one-shot reasoning (fast thinking)*, and *stepwise reasoning (slow thinking)*. Although all agents can answer all queries, each agent has different advantages and disadvantages, enabling hyper agent to choose the optimal transition and re-route on failure. The implementation details of agents are shown in the supplementary material (Appendix B).

**Shared Memory.** All agents read from and write to a structured *shared memory* $m_t$ at the $t$-th step that accumulates intermediate variables, perception results, program history, verifier feedback, and task metadata. We keep the formalism minimal: when an agent runs for one cycle, it appends its new memory $\Delta m_t$, and $m_{t+1} = m_t \cup \Delta m_t$. Memory is append-only so the full reasoning process is auditable and visible to the hyper agent.

**Execution Step.** At step $t$ the system is in $(s_t, m_t)$. The hyper-agent observes the memory $m_t$ and selects the next state $s_{t+1}$ via the learned transition function $\delta_\theta$:

$$s_{t+1} = \delta_\theta(s_t, m_t), \quad s_{t+1} \in S. \tag{1}$$

If $s_{t+1} \in S_{\text{agent}}$, the corresponding agent executes its rule-based sub-automaton until returning to the hyper automaton and updating the memory; if $s_{t+1} = $ FINAL or $t > T$ where $T$ is the max step limit, the episode terminates.

**Output.** The answer space $\Lambda$ contains the required output $\hat{y}$ for visual reasoning. For example, $\Lambda = \{y \mid y \text{ is text for VQA, bounding box for VG, etc}\}$. The output function $\Gamma$ extracts the output from the memory $m_t$ at FINAL state: $\hat{y} = \Gamma(\text{FINAL}, m_t)$.

### 3.3 TRAINABLE TRANSITION FUNCTION (HYPER AGENT)

The transition function $\delta_\theta$ in Equation 1 is implemented by a trainable LLM-based *hyper agent* $\mathcal{F}_\theta$. This agent acts as the state-transition controller, selecting the next state $s_{t+1}$ from a limited set of available candidate states. Since the LLM requires textual input, we derive a prompt $x_t$ from the shared memory $m_t$. The template for constructing $x_t$ from $m_t$ is shown below:

---
**Prompt 3.1: LLM State Controller in Hyper Agent**

You are an AI assistant to control the state of a multi-step visual reasoning system. Your task is to decide the next state the system should transition to based on the current state and history.
<TaskDescription>{task_title}{task_description}</TaskDescription>
<Query>{query}</Query>
<Feedback>{feedback}</Feedback>
{code}
<Variables>{variables}</Variables>
<StateHistory>{state_history}</StateHistory>

---

> <State>{state}</State>
> <CurrentStep>{current_step}</CurrentStep>
> Based on the information above, determine the next state the system should transition to. Choose from the following states:
> <StateCandidates>{next_state_candidates}</StateCandidates>
> Return the name wrapped in <NextState> tags.

Our hyper agent $\mathcal{F}_\theta$ maps the prompt $x_t$ to a distribution over the available states, from which $s_{t+1}$ is selected, either through greedy decoding or stochastic sampling.

The parameter $\theta$ of the hyper agent is supervised finetuned (SFT) on our collected transition trajectory dataset $\mathcal{D}$ (subsection 3.4). Each training example provides a textual memory $x_t$ as prompt and a target next state chosen by scanning branches in the trajectory tree that lead to successful and higher final scores:

$$\theta \leftarrow \arg \min_\theta \mathcal{L}_{\text{SFT}}(\theta; \mathcal{D}) \tag{2}$$

This objective guides the hyper agent on how to switch between sub-agents, and finalize the output.

### 3.4 DATASET GENERATION

Learning the transition policy of the hyper automaton requires examples of how agent states interact during visual reasoning. We therefore build a dataset of transition trajectories. We regard the set of possible transition trajectories from an initial state as a trajectory tree $\mathcal{T}(v, q)$ (Kearns et al., 2002) that records, for each node: the state history, intermediate reasoning outcomes, and final metric scores, as a textual prompt $x_t$ based on prompt 3.1. We collect this data by running MATA while systematically traversing each next-state option rather than committing to a single path. Unlike end-to-end LLM/VLM training, this procedure explicitly explores the space of possible agent states and yields labeled decisions for our model.

Concretely, we sample images and queries from the training splits of GQA (Hudson & Manning, 2019), OK-VQA (Marino et al., 2019), and RefCOCO/RefCOCO+/RefCOCOg (Kazemzadeh et al., 2014) and run the hyper automaton $\mathcal{M}_\theta$ step-wise. Rather than limiting to a single route, we expand a bounded trajectory tree to depth $T$: at each node (state) the controller branches over the possible next states $s_{t+1} \in S$, executes the corresponding sub-automaton, and saves a memory checkpoint $m_{t+1}$. When a terminal state is reached (e.g., FINAL), which by construction corresponds to a *leaf* of the tree $\mathcal{T}$, the output function $\Gamma$ produces a prediction $\hat{y}$ for the given image-query pair $(v, q)$ with ground truth $y$. We then compute a scalar task score for that leaf: for VG we use $\text{IoU}(\hat{y}, y)$; for VQA we use $\text{Acc}(\hat{y}, y)$. During data collection we perform a near-exhaustive expansion of the transition tree to a fixed depth, which is tractable with the current three agents but, we acknowledge, grows rapidly as more agents/states are added.

**Bottom-up node scoring.** As a result, each leaf node $s \in \text{Leaves}(\mathcal{T})$ is associated with a prediction $\hat{y}_s$ and ground truth $y$, from which we compute a scalar score. We assign values to all nodes by propagating these scores upward from the leaves:

$$V(s) \triangleq \begin{cases} \text{metric}(\hat{y}_s, y), & s \in \text{Leaves}(\mathcal{T}), \\ \max_{s' \in \text{Child}(s)} V(s'), & \text{otherwise.} \end{cases} \tag{3}$$

To train the LLM state controller, we convert each multi-choice transition into supervised examples. For every decision point at state $s_t$ with corresponding textual prompt $x_t$, we determine the optimal next state $s_t^\star$ by selecting the child node that leads to the subtree with the highest propagated value. Formally, for a state $s_t$ with its set of next states $\text{Child}(s_t) \subseteq S$, we choose:

$$s_t^\star \in \arg \max_{s \in \text{Child}(s_t)} V(s). \tag{4}$$

The chosen state $s_t^\star$ becomes the label for the corresponding node prompt $x_t$, and together they form a training example. Repeating this over all decision points produces a dataset of message histories paired with optimal next states, $\mathcal{D} = \{(x_i, s_i^\star)\}_{i=1}^N$. Finally, we reformat the collected examples into instruction-completion pairs suitable for supervised finetuning of LLM. Training on this dataset enables the model to learn how to control the transitions of a hyper automaton. In total, we build

the SFT dataset MATA-SFT-90K containing $N = 90{,}854$ examples. We show the data example in Appendix H.

## 3.5 INFERENCE

Given an image-query pair $(v, q)$, we initialize the shared memory $m_0$ and enter the initial state $s_0 = $ INITIAL. At step $t$, the hyper agent $\mathcal{F}_\theta$ reads the current context $x_t$ and selects the next state $s_{t+1}$ using the learned transition in Equation 1. If $s_{t+1} \in S_{\text{agent}}$, the corresponding sub-agent executes one cycle of its rule-based sub-automaton, appends its intermediate result to memory, and returns to the hyper automaton. If $s_{t+1} = $ FAILURE, this state indicates that the selected agent $s_t$ reports an unrecoverable error and the system will invoke the hyper agent to choose a new state $s_{t+1}$ while temporarily removing the failed agent $s_t$ from the state candidates to avoid infinite retries. If $s_{t+1} = $ FINAL or the step $t$ exceeds the limit $T$, the system terminates and returns the final result $\hat{y}$.

## 4 EXPERIMENTS AND RESULTS

**Implementation Details.** We implement MATA in PyTorch (Paszke et al., 2019) and conduct all experiments on 4 RTX 4090 GPUs. The system uses interchangeable foundation models; unless otherwise stated we adopt InternVL2.5 (8B) (Chen et al., 2025) as the VLM, Florence2-L (Xiao et al., 2024) for object detection, DepthAnythingV2 (Yang et al., 2024) for depth, and a Qwen3 (4B) (Yang et al., 2025) LLM for the trainable state controller in the hyper agent. The LLM is supervised finetuned on MATA-SFT-90K using AdamW, cosine decay with 5% warm-up, global batch size 64, for 8 epochs; decoding is guided at inference to ensure the output format. As MATA-SFT-90K is a dataset collected by running our pipeline on multiple source datasets, "training on dataset X" means training on the subset of MATA-SFT-90K whose trajectories were generated from the training split of X. We use three SFT configurations for the hyper agent: (i) domain-specific: trained on the training split of the target dataset and evaluated on its test split; (ii) domain-transfer[1]: trained on the dataset which is not the target dataset for evaluation; and (iii) general: trained jointly on the whole dataset. We follow the official splits of all the benchmark datasets, reporting accuracy. For fairness, when comparing with compositional baselines we keep the same foundation models, and for monolithic models we use the available public checkpoints with their official code. In the inference, we limit the max step of MATA $T = 15$ to avoid infinite running. The prompt template is shown in the Appendix G in supplementary material.

**Evaluation Protocol.** We evaluate on GQA (Hudson & Manning, 2019), OK-VQA (Marino et al., 2019), RefCOCO/RefCOCO+/RefCOCOg (Kazemzadeh et al., 2014), and Ref-Adv (Akula et al., 2020) following the previous works (Surís et al., 2023; Ke et al., 2024; Cai et al., 2025), with accuracy as the metric. We compare against the previous compositional methods which are training-required (Khan et al., 2024; Ke et al., 2025a) or training-free (Surís et al., 2023; Ke et al., 2024; Cai et al., 2025), and monolithic methods (Li et al., 2023b; Zhu et al., 2023; Liu et al., 2023a; Su et al., 2023; Han et al., 2023; Dai et al., 2023; Li et al., 2023a; Wang et al., 2024; Bai et al., 2025b; Chen et al., 2025; Zhu et al., 2025; Wang et al., 2025b; OpenAI, 2024; Tiong et al., 2022; Yang et al., 2022; Alayrac et al., 2022).

## 4.1 QUANTITATIVE RESULTS

**Compositional Image Question Answering.** On GQA (Hudson & Manning, 2019), which emphasizes complex compositional reasoning over spatial relations and attributes, MATA reaches 64.9% accuracy (Table 2), surpassing previous trainable compositional methods HYDRA and VisRep, training-free baselines such as ViperGPT. It is also competitive with strong monolithic VLMs, exceeding InternVL3.5 and Qwen2.5-VL. The gains stem from the learned transition policy, and the hyper agent understands the capacity of agents. Easy queries invoke SPECIALIZED perception first and escalate to ONESHOT or STEPWISE only on failure or low confidence, whereas difficult cases route directly to STEPWISE to maximize the reasoning. When the range of data is narrow and distinctive, the domain-specific setting can calibrate priors more precisely; when compositional

---

[1] Our *domain-transfer* term is scoped to the hyper agent: it is trained on non-test-dataset transition trajectories, and never sees the optimal trajectories in other datasets.

Table 2: **Performance on GQA dataset.**

| Type | | Method | Acc. |
|---|---|---|---|
| Monolithic | • | BLIP-2 (Li et al., 2023b) | 45.5 |
| | • | MiniGPT-4 (13B) (Zhu et al., 2023) | 30.8 |
| | • | LLaVA (13B) (Liu et al., 2023a) | 41.3 |
| | • | PandaGPT (13B) (Su et al., 2023) | 41.6 |
| | • | ImageBind-LLM (7B) (Han et al., 2023) | 41.2 |
| | • | InstructBLIP (13B) (Dai et al., 2023) | 49.5 |
| | • | Otter (7B) (Li et al., 2023a) | 50.0 |
| | • | Qwen2-VL (7B) (Wang et al., 2024) | 34.3 |
| | • | Qwen2.5-VL (7B) (Bai et al., 2025b) | 62.4 |
| | • | Qwen3-VL (4B) (Bai et al., 2025a) | 51.6 |
| | • | InternVL2.5 (8B) (Chen et al., 2025) | 61.5 |
| | • | InternVL3 (8B) (Zhu et al., 2025) | 62.4 |
| | • | InternVL3.5 (8B) (Wang et al., 2025b) | 63.8 |
| | • | GPT-4o-2024-05-13 (OpenAI, 2024) | 58.5 |
| Compositional | ○ | IdealGPT (You et al., 2023) | 41.7 |
| | • | ViperGPT (Surís et al., 2023) | 37.9 |
| | • | VisRep (Khan et al., 2024) | 51.4 |
| | ○ | HYDRA (Ke et al., 2024) | 52.8 |
| | ◎ | MATA (Ours) (General) | **64.9** |
| | ◎ | MATA (Ours) (Domain-Specific) | 64.7 |

Table 3: **Performance on OK-VQA dataset.**

| Type | | Method | Acc. |
|---|---|---|---|
| Monolithic | • | PNP-VQA (Tiong et al., 2022) | 35.9 |
| | • | PICa (Yang et al., 2022) | 43.3 |
| | • | BLIP-2 (Li et al., 2023b) | 45.9 |
| | • | Flamingo (9B) (Alayrac et al., 2022) | 44.7 |
| | • | MiniGPT-4 (13B) (Zhu et al., 2023) | 37.5 |
| | • | LLaVA (13B) (Liu et al., 2023a) | 42.5 |
| | • | InstructBLIP (13B) (Dai et al., 2023) | 47.9 |
| | • | Qwen2-VL (7B) (Wang et al., 2024) | 28.3 |
| | • | Qwen2.5-VL (7B) (Bai et al., 2025b) | 71.8 |
| | • | Qwen3-VL (4B) (Bai et al., 2025a) | 44.4 |
| | • | InternVL2.5 (8B) (Chen et al., 2025) | 75.2 |
| | • | InternVL3 (8B) (Zhu et al., 2025) | 74.7 |
| | • | InternVL3.5 (8B) (Wang et al., 2025b) | 75.7 |
| | • | GPT-4o-2024-05-13 (OpenAI, 2024) | 33.4 |
| Compositional | ○ | IdealGPT (You et al., 2023) | 19.4 |
| | • | ViperGPT (Surís et al., 2023) | 40.7 |
| | • | VisRep (Khan et al., 2024) | 46.7 |
| | ○ | HYDRA (Ke et al., 2024) | 59.4 |
| | ○ | DWIM (Ke et al., 2025a) | 62.8 |
| | ◎ | MATA (Ours) (General) | 76.0 |
| | ◎ | MATA (Ours) (Domain-Specific) | **76.5** |

Agentic types: • non-agentic/non-specified; ○ single-agent; ◎ multi-agent.

Table 4: **Quantitative comparison (accuracy) on referring expression comprehension task** on RefCOCO, RefCOCO+, RefCOCOg (Kazemzadeh et al., 2014) and Ref-Adv (Akula et al., 2020) set. Note there is no training set in Ref-Adv, so all scores are domain-transfer.

| Type | | Method | RefCOCO | RefCOCO+ | RefCOCOg | Ref-Adv |
|---|---|---|---|---|---|---|
| Monolithic | • | GLIP-L (Li et al., 2022) | 55.0 | 51.1 | 54.6 | 55.7 |
| | • | KOSMOS-2 (Peng et al., 2023) | 57.4 | 50.7 | 61.7 | - |
| | • | YOLO-World-X (Cheng et al., 2024) | 12.1 | 12.1 | 32.9 | 32.2 |
| | • | YOLO-World-V2-X (Cheng et al., 2024) | 19.8 | 16.8 | 36.5 | 33.1 |
| | • | GroundingDINO-T (Liu et al., 2023b) | 61.6 | 59.7 | 60.6 | 60.5 |
| | • | GroundingDINO-B (Liu et al., 2023b) | 90.8 | 84.6 | 80.3 | 78.0 |
| | • | SimVG (Dai et al., 2024) | 94.9 | 91.0 | 88.9 | 74.4 |
| | • | Florence2-B (Xiao et al., 2024) | 94.5 | 91.2 | 88.3 | 72.2 |
| | • | Florence2-L (Xiao et al., 2024) | 95.1 | 92.5 | 90.9 | 71.8 |
| | • | GPT-4o-2024-05-13 (OpenAI, 2024) | 30.5 | 26.2 | - | - |
| | • | Qwen2.5-VL-72B (Bai et al., 2025b) | 94.6 | 92.2 | 90.3 | - |
| Compositional | • | Code-bison (Stanić et al., 2024) | 44.4 | 38.2 | - | - |
| | • | ViperGPT (Surís et al., 2023) | 68.6 | 73.8 | 68.7 | 58.2 |
| | • | VisRep (Khan et al., 2024) | 55.2 | 51.1 | - | - |
| | ○ | HYDRA (Ke et al., 2024) | 65.7 | 66.2 | 59.9 | 48.3 |
| | ○ | DWIM (Ke et al., 2025a) | 82.7 | 74.2 | - | - |
| | ○ | NAVER (Cai et al., 2025) | 96.2 | 92.8 | 91.6 | 75.4 |
| | ◎ | MATA (Ours) (General) | 96.3 | 93.8 | 90.7 | **77.3** |
| | ◎ | MATA (Ours) (Domain-Specific) | **96.3** | **93.9** | **90.8** | - |

Agentic types: • non-agentic/non-specified; ○ single-agent; ◎ multi-agent.

patterns are shared across sources, joint training (general) regularizes transitions and reduces over-fitting. In GQA we observe the latter, many patterns appear across sources in MATA-SFT-90K, so the general setting achieves better performance.

**External Knowledge-Dependent Image Question Answering.** On OK-VQA (Marino et al., 2019), which requires external knowledge, MATA achieves **76.5%** accuracy (Table 3), surpassing prior compositional systems such as DWIM (62.8%) and HYDRA (59.4%), respectively, and outperforming recent monolithic VLMs including Qwen2.5-VL (71.8%) and InternVL3.5 (75.7%). Gains come from the learned hyper agent transition: for easy queries the hyper agent first invokes SPECIALIZED perception and escalates to the STEPWISE or ONESHOT reasoner only on failure or

Table 5: **Ablation of hyper agent.** In this table, we report the accuracy for all VQA and referring expression comprehension benchmarks, and the inference time per query (tested on GQA). *HA: Hyper Automaton. Transition: Transition policy ($\delta_\theta$). SFT: Supervised finetuning.* Refer to subsection 4.2 for details.

| Components | | | Accuracy (↑) | | | | | | | Time (↓) |
|---|---|---|---|---|---|---|---|---|---|---|
| HA | Transition | SFT | GQA | OK-VQA | RefCOCO | RefCOCO+ | RefCOCOg | Ref-Adv | | Avg Sec. |
| ✗ | Exhaustive | ✗ | 57.7 | 71.5 | 87.7 | 85.6 | 81.7 | 73.1 | | 34.58 |
| ✓ | Random | ✗ | 57.1 | 71.1 | 85.3 | 83.8 | 81.1 | 73.2 | | 6.91 |
| ✓ | LLM | ✗ | 58.5 | 75.1 | 95.8 | 93.5 | 88.0 | 76.0 | | 8.07 |
| ✓ | LLM | ✓ | **64.9** | **76.5** | **96.3** | **93.9** | **90.8** | **77.3** | | **8.01** |

Table 6: **Generalizability results.** The top-left header cell uses a diagonal split to indicate *Training Data* (rows, ↓) versus *Test Data* (columns, →). Diagonal values (domain-specific) train and test on the *same* dataset; off-diagonal values evaluate cross-domain/task transfer (domain-transfer) . The last row reports joint training on the whole MATA-SFT-90K dataset (general) . Off-diagonal values are close to the diagonal ones, indicating strong generalizability of the learned transition policy.

| Training \ Test | VQA | | Visual Grounding | | | |
|---|---|---|---|---|---|---|
| | GQA | OK-VQA | RefCOCO | RefCOCO+ | RefCOCOg | Ref-Adv |
| GQA | 64.7 | 75.8 | 96.1 | 93.7 | 90.4 | 77.0 |
| OK-VQA | 64.1 | 76.5 | 96.2 | 93.8 | 90.5 | 76.9 |
| RefCOCO | 63.8 | 75.5 | 96.3 | 93.9 | 90.8 | 77.2 |
| RefCOCO+ | 63.6 | 75.4 | 96.2 | 93.9 | 90.7 | 77.1 |
| RefCOCOg | 63.1 | 75.4 | 96.1 | 93.7 | 90.8 | 77.2 |
| All | 64.9 | 76.0 | 96.3 | 93.8 | 90.7 | 77.3 |

low confidence; for difficult queries it directly selects STEPWISE for multi-step reasoning, with competitive re-entry into SPECIALIZED or ONESHOT to reason combining the previous findings and new evidence. We observe the domain-specific setting holds a small edge, likely because of the narrow diversity of the reasoning pattern required in the dataset, whereas joint training (general) slightly dilutes these knowledge.

**Referring Expression Comprehension.** On popular benchmarks RefCOCO, RefCOCO+, RefCOCOg (Kazemzadeh et al., 2014) and Ref-Adv (Akula et al., 2020), MATA obtains state-of-the-art performance (Table 4). It sets a new state-of-the-art on these datasets, exceeding strong monolithic and compositional baselines. Notably, Ref-Adv only contains a test set, which means the MATA-SFT-90K does not contain the data collected from it, showing promising domain-transfer generalizability of MATA. Note that due to learned transition, short simple queries are solved by SPECIALIZED perception with verification, while complex cases trigger STEPWISE and ONESHOT reasoning. Domain-specific SFT is slightly stronger because the language query styles is dataset-specific.

## 4.2 ABLATION STUDIES

**Hyper Agent.** Table 5 isolates the main contribution of the trainable hyper agent and the hierarchical automaton design. We compare: (1) **Exhaustive Ensemble** without hierarchical automaton (HA): exhaustively call all sub-agents and aggregate with a VLM; (2) **Random Transition**: HA enabled but the next state is chosen randomly; (3) **LLM without SFT:** a pretrained LLM is used as the state controller (no supervised finetuning); (4) **LLM + SFT:** a supervised finetuned LLM controls transitions. Both the exhaustive baseline and random transition yield the weakest performance, but introducing the hyper automaton already cuts runtime significantly. Replacing random with a pretrained LLM in hyper agent improves accuracy across tasks. This suggests that (i) the hyper automaton and the LLM primarily drive effective multi-agent collaboration and competition and (ii) SFT further helps the understanding of the capacity of agents in different types of questions.

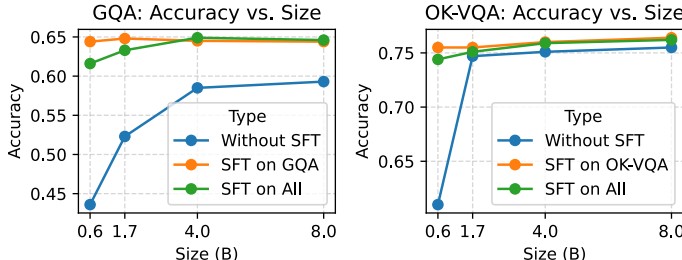 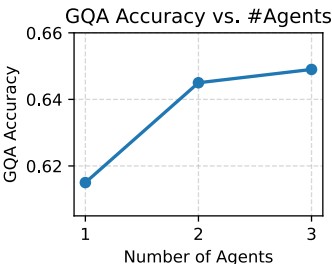

Figure 3: **Results of different LLM sizes.** Accuracy versus the model size (in billions of parameters) of the hyper agent's LLM state controller. Left: GQA; right: OK-VQA. X-axis: LLM size; Y-axis: accuracy.

Figure 4: **Results of different numbers of sub-agents.** X-axis: number of sub-agents; Y-axis: accuracy in GQA.

**Generalizability.** We conduct generalization analysis by training the hyper agent on GQA subset only of MATA-SFT-90K dataset, OK-VQA subset only, or the whole dataset. Table 6 organizes results by different training/evaluation types: domain-specific, domain-transfer, and general. domain-transfer performance is strong in both directions (GQA→OK-VQA; OK-VQA→GQA) with less than 1% difference. The model trained on all data reaches similar performance to the model trained on the corresponding subset only, indicating the controller learns a task-agnostic transition policy with minimal negative impact. We further discuss the effects in the next paragraph.

**LLM Size.** Figure 3 compares the sizes of the LLM state controller from 0.6B to 8B under three settings: (i) no SFT, (ii) domain-specific SFT, and (iii) SFT on all. With domain-specific SFT, even small models (0.6B/1.7B) perform competitively matching 4B and 8B. When finetuned jointly on all data, small models are worse than 4B/8B by a few percentage points, indicating limited capacity to absorb cross-task knowledge. Without SFT, accuracy drops sharply for smaller models and improves mainly with size. Balancing accuracy and efficiency, we choose 4B as default, as it produces near-optimal results with substantially lower memory, while larger models yielding only marginal gains.

**Number of Agents.** We ablate the number of agent states to quantify benefits beyond our 3-agent design. On GQA, a single *Specialized* agent reaches 61.5%, adding the *Oneshot* reasoner lifts accuracy to 64.5%, and adding the *Stepwise* reasoner yields a marginal further gain to 64.9% (Figure 4). The small improvement from 2 to 3 agents indicates diminishing improvements on current benchmarks, suggesting that the agent count is not the major factor. We therefore use three agents in MATA.

**More Analysis.** We discuss more analysis for generalizability in Appendix C, hyper agent in Appendix D, efficiency in Appendix E, comparison with direct SFT in Appendix F, and the qualitative examples in Appendix I in supplementary materials.

## 5 CONCLUSION

We present MATA, a visual reasoning method that uses a trainable hyper agent to learn the transition policy of a hierarchical finite-state automaton. By transitioning between agents based on a shared memory, the system reduces hallucinations, and preserves explainability through explicit states and context. To supervise the hyper agent, we introduced the transition-trajectory dataset MATA-SFT-90K, which converts the trajectory data into a standard SFT format and adapts as agents are added. From experiments, MATA achieves state-of-the-art performance across multiple datasets. **Limitations.** The data generation pipeline performs a near-exhaustive transition search over the state space; this is tractable with the current three agents but may become costly as the number of states grows.

ACKNOWLEDGMENTS

This research is sponsored by the DARPA Assured Neuro Symbolic Learning and Reasoning (ANSR) program under award number FA8750-23-2-1016.

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
