## A    THE USE OF LARGE LANGUAGE MODELS

We declare that LLMs (GPT-5/5.1/5.2) are used for the paper language polishing.

## B    IMPLEMENTATION OF AGENTS

In this section we introduce the detailed implementation of the four agents shown in Figure 5. The hyper agent is triggered at each decision point of the hyper automaton (on INITIAL, and after any agent returns) then summarizes the shared memory $m_t$ and applies the learned transition $\delta_\theta$ to select the next state $s_{t+1}$. Other agents are triggered only when selected by the hyper agent. Upon entry, the selected agent always starts at its internal INITIAL state; reaching the agent's RETURN state hands control back to the hyper automaton.

We implemented *three* agents to span levels of reasoning: a *Specialized* System-1 perception agent, an *Oneshot* fast thinking agent, and a *Stepwise* slow thinking agent. Each agent brings different trade-offs. The *Specialized* agent is fast and verifiable for easier subtasks such as finding an object without complex relations, but lacks depth for multi-step compositional reasoning. The *Oneshot* reasoner is cheap and effective on moderately compositional queries, yet might fail on edge cases because it generates the full workflow without accessing the intermediate variable in the workflow. The *Stepwise* agent is designed for complex reasoning via verified program execution, but incurs higher latency and cost. The formulation is modular and scales to additional agents without changing the other part of the system.

### B.1    HYPER AGENT

As illustrated in Figure 5 (top-left), the hyper agent is triggered from hyper automaton, uses the *Memory Prompter* to convert the current shared memory snapshot $m_t$ into a text prompt $x_t$, and feeds it to the trainable *LLM State Controller* to propose the next state. If the LLM fails to generate a valid proposal, we re-prompt once with extra feedback. The output is then checked by the *Transition Verifier*, which enforces valid state selection. On success, the hyper agent returns the chosen state to the hyper automaton and appends the decision to $m_t$.

### B.2    ONESHOT REASONER

In Figure 5 (top-right), the oneshot reasoner enters at *Initial* from the hyper automaton, calls the *Code Generator* to produce a Python program, and passes it to the *Code Verifier* for format checks; generation or validation failure triggers a regeneration. Verified code is executed by the *Code Interpreter* in a Python environment; runtime errors trigger regeneration with extra feedback. On success, the program, execution history, and feedback are appended to $m_t$ and the agent returns to the hyper

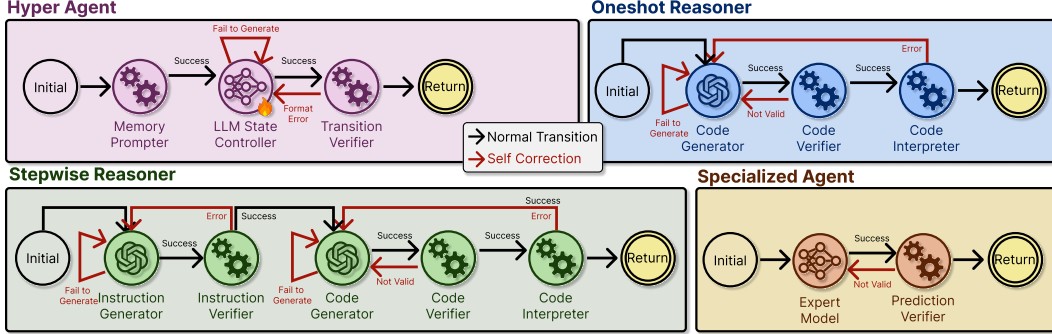

Figure 5: **Details of agents in MATA.** Each block shows the sub-automaton executed when the hyper automaton transits into that agent. Black arrows indicate the normal paths; red arrows show local error-correction paths. Persistent failures transition to FAILURE state of the hyper automaton (omitted for clarity).

Table 7: **More generalizability results.** The top-left header cell uses a diagonal split to indicate *Training Data* (rows, ↓) versus *Test Data* (columns, →). Row *Single the same dataset* trains each LLM state controller in hyper agent on each training set of the dataset and tests on the test set of the same dataset (domain-specific) ; row *All exclude the dataset* trains on the union of the remaining datasets and tests on the held-out column dataset (domain-transfer) ; row *All include the dataset* trains jointly on all datasets (general) . Off-domain accuracies are close to the domain-specific ones, indicating that the learned transition policy generalizes across tasks.

| Training \ Test | VQA | | Visual Grounding | | |
|---|---|---|---|---|---|
| | GQA | OK-VQA | RefCOCO | RefCOCO+ | RefCOCOg |
| Single the same dataset | 64.7 | 76.5 | 96.3 | 93.9 | 90.8 |
| All exclude the dataset | 63.5 | 75.4 | 96.1 | 93.8 | 90.7 |
| All include the dataset | 64.9 | 76.0 | 96.3 | 93.8 | 90.7 |

automaton; if verification or execution repeatedly fails, the agent triggers FAILURE, and returns to the hyper automaton.

### B.3 STEPWISE REASONER

The stepwise reasoner (Figure 5, bottom-left) handles more complex, slower reasoning: from hyper automaton (*Initial*), the *Instruction Generator* proposes the next one-step plan based on $m_t$, which the *Instruction Verifier* validates; a verification error or failure to generate triggers one regeneration. The accepted plan is translated by the *Code Generator*, checked by the *Code Verifier*, and executed by the *Code Interpreter*; each stage includes error-correction loops as annotated in the figure. If execution succeeds, new context (variables, history, feedback) is written into $m_t$ and the agent returns to the hyper automaton; if any stage stays invalid after multiple attempts, the agent triggers FAILURE and returns control to the hyper automaton.

### B.4 SPECIALIZED AGENT

As shown in Figure 5 (bottom-right), the specialized agent begins from hyper automaton, runs an *Expert Model* (e.g., VLM, object detector), and its output is verified by the *Prediction Verifier* for extra checks. If the output is not valid, the agent performs one adaptive retry; otherwise it commits the intermediate results and verifier feedback to $m_t$ and returns to the hyper automaton. Persistent invalid results trigger failure and return control to the hyper automaton.

## C MORE ANALYSIS FOR GENERALIZABILITY

We conducted further generalization analysis by training the hyper agent with several more dataset configurations. As shown in Table 7, we classify the results with three different training data configurations: *Single the same dataset* means that the hyper agent is trained on the training set of the test dataset. *All exclude the dataset* means the hyper agent is trained on the whole MATA-SFT-90K dataset but excluding the corresponding training data from the same dataset, to ensure that it is domain-transfer. *All include the dataset* means the hyper agent is trained on the whole MATA-SFT-90K dataset which includes the training data from the same dataset to be evaluated. From the extra results, the observation further supports our findings in section 4.2.

Across all benchmarks, the domain-transfer setting (*All exclude the dataset*) is around 1 percentage of the domain-specific setting (*Single the same dataset*). The general model jointly training on the whole dataset (*All include the dataset*) reaches similar performance of domain-specific. The small gaps indicate that the learned transition policy is largely task-agnostic: it transfers across VQA and grounding without per-dataset tuning, and gains from multi-dataset SFT do not harm in-domain accuracy. Practically, this suggests a single hyper agent can be trained once and reused across visual reasoning tasks.

## D    MORE ANALYSIS FOR HYPER AGENT

We conduct experiments (Table 8) using different models as the state controller in hyper agent. We trained the Qwen3-4B (LLM) (Yang et al., 2025) and Qwen3-VL 4B (VLM) (Bai et al., 2025a) on the full set of MATA-SFT-90K. From the results, the state controller is insensitive to the backbone type: swapping the LLM for a VLM yields near-similar performance across all datasets. We therefore adopt the LLM controller to minimize system complexity and resource requirements while retaining performance.

Table 8: **More results for the state controller model of hyper agent.** All models are trained on the trajectory transitions of the full MATA-SFT-90K.

| State Controller | GQA | OK-VQA | RefCOCO | RefCOCO+ | RefCOCOg | Ref-Adv |
|---|---|---|---|---|---|---|
| Qwen3 (4B) (LLM) | 64.9 | 76.0 | 96.3 | 93.9 | 90.8 | 77.3 |
| Qwen3-VL (4B) (VLM) | 64.6 | 76.3 | 96.4 | 94.0 | 89.3 | 76.9 |

## E    MORE ANALYSIS FOR EFFICIENCY

To further analyze the system time and spatial complexity, we collected and calculated the inference time (seconds), LLM API costs (USD, GPT-4o mini) and vRAM usage (GB) per query, between the state-of-the-art monolithic method Qwen2.5-VL (72B) (Bai et al., 2025b) with 4-bit quantization, the open sourced compositional agentic method HYDRA (Ke et al., 2024), the baseline which call all sub-agents exhaustively to aggregate the final answer, and the proposed method MATA. All measurements are taken on a single NVIDIA L40s 48GB GPU on RefCOCO dataset. As shown in Table 9, MATA attains the best efficiency compared with HYDRA and exhaustive baseline and is comparable to the monolithic baseline, while achieving the lowest API cost and a moderate vRAM usage (substantially below the 72B model).

Table 9: **More analysis for efficiency.** We compare the inference time in seconds, LLM API costs in USD, and vRAM in GB, on RefCOCO dataset.

| Method | Time (Seconds) | LLM API Cost (USD) | vRAM Usage (GB) |
|---|---|---|---|
| Qwen2.5-VL (72B) | 6.27 | – | 43.70 |
| HYDRA | 14.93 | 0.00332 | 17.59 |
| Exhaustive | 32.74 | 0.00531 | 13.08 |
| MATA | 6.10 | 0.00069 | 19.64 |

## F    COMPARISON WITH DIRECT SFT

We compare two paradigms: (i) *direct SFT* of a single VLM (Qwen3-VL (4B) (Bai et al., 2025a), InternVL2.5 (8B) (Chen et al., 2025)) to output answers, and (ii) *MATA*, which finetunes only the hyper agent's state controller model as a transition policy. As summarized in Table 10, answer-only direct SFT can improve in-domain accuracy but often harms cross-task generalization, which is consistent with catastrophic forgetting of the model's latent "think-then-answer" ability, while MATA maintains strong transfer because it learns transitions between agents rather than a direct monolithic question-to-answer mapping. The direct SFT on the related dataset gains for Qwen3-VL (4B) reflect its lower zero-shot starting point; stronger VLM like InternVL2.5 (8B) is typically harder to improve via answer-only direct SFT. Overall, MATA delivers higher accuracy and more robust cross-task performance than direct SFT.

## G    PROMPT TEMPLATES

MATA uses LLMs in multiple places, including: (1) A trainable *LLM state controller* in the *Hyper Agent* routes between states by reading a summarized snapshot of the shared memory. (2) An *Instruction Generator* in *Stepwise Reasoner* proposes the next micro-plan. (3) A *Code Generator* in

Table 10: **Direct SFT vs. MATA.** We compare (i) directly finetuning a VLM baseline (Qwen3-VL 4B, InternVL-2.5 8B) to output answers and (ii) MATA (SFT on hyper agent) on GQA and OK-VQA. All values are accuracy (%). "–" denotes using public weights without task-specific finetuning. *Training Dataset* indicates which task split was used for SFT. Color codes follow prior tables: domain-specific, domain-transfer, general. Note the pretraining data of LLM/VLMs are unknown; colors are for ease of comparison.

| Method | Training Dataset | Test Dataset | |
| --- | --- | --- | --- |
| | | GQA | OK-VQA |
| Qwen3-VL (4B) | – | 51.6 | 44.4 |
| Qwen3-VL (4B) | GQA | 63.2 | 61.6 |
| Qwen3-VL (4B) | OK-VQA | 57.5 | 68.1 |
| Qwen3-VL (4B) | GQA, OK-VQA | 63.1 | 66.9 |
| InternVL2.5 (8B) | – | 61.5 | 75.2 |
| InternVL2.5 (8B) | GQA | 63.9 | 64.1 |
| InternVL2.5 (8B) | OK-VQA | 35.8 | 72.4 |
| InternVL2.5 (8B) | GQA, OK-VQA | 63.8 | 65.2 |
| MATA | – | 58.5 | 75.1 |
| MATA | GQA | 64.7 | 75.8 |
| MATA | OK-VQA | 64.1 | 76.5 |
| MATA | GQA, OK-VQA | 64.9 | 76.4 |

*Stepwise Reasoner* generates Python code for that step. (4) The *Oneshot Reasoner* employs another *Code Generator* to produce a single-pass program. Across roles, prompts are concise, instruction-style templates that expose the relevant slice of shared memory and tool signatures and require outputs in strict JSON/XML blocks for reliable parsing. The prompt template of the *LLM state controller* is shown in prompt 3.1. The following prompt blocks show detailed prompt templates of the LLMs.

---

**Prompt G.1: Instruction Generator in Stepwise Reasoner**

You are an AI assistant designed to assist with compositional visual reasoning tasks providing valid step by step instruction for answering questions and understanding visual information.
Instruction Settings
———————
<InstructionSetting>{instruction_setting}</InstructionSetting>
Skills Overview
——————
The following are the skills that you can use to solve the query:
<Skills>{skills}</Skills>
Task Description
——————-
Review the task description below to understand the problem context:
<TaskDescription>{task_title}{task_description}</TaskDescription>
Example Instructions
———————-
How to Use these skills:
<Examples>{instruction_example}</Examples>
User Query
————-
This is the query you need to solve:
<Query>{query}</Query>
Current Step
————
<Step>{current_step}</Step>
Previous Instructions
————————
<PreviousInstructions>{previous_instructions}</PreviousInstructions>
Previously Executed Code

---

—————————
<ExecutedCode>{previous_code}</ExecutedCode>
Execution Results
—————
<ExecutionResults>{execution_results}</ExecutionResults>
Available Variables
——————-
<Variables>{variables_info}</Variables>
——————-
Based on the current context, generate possible next instructions to help solve the query. For each instruction, assign a probability score indicating how promising it will lead to the final answer.
Your response must be in this JSON array format:
{"instructions": [ {"instruction": "specific instruction", "probability": 0.X}, {"instruction": "another instruction", "probability": 0.Y}, ... ]}

---

### Prompt G.2: Code Generator in Stepwise Reasoner

You are a helpful assistant specializing in visual reasoning tasks. Your goal is to generate Python code that solves a visual reasoning query using the provided code API and examples.
API Specification
——————
Use the following code API to guide your solution:
<CodeAPI>{code_api}</CodeAPI>
Task Description
——————-
Review the task description below to understand the problem context:
<TaskDescription>{task_title}{task_description}</TaskDescription>
Example Code
————
Here is an example that illustrates the expected format and approach:
<Examples>{code_example}</Examples>
User Query
———-
This is the query you need to solve:
<Query>{query}</Query>
Current Step
————
<Step>{current_step}</Step>
Previous Instructions
———————
<PreviousInstructions>{previous_instructions}</PreviousInstructions>
Current Instruction
——————-
<Instruction>{instruction}</Instruction>
Previously Executed Code
————————
<ExecutedCode>{previous_code}</ExecutedCode>
Execution Results
——————-
<ExecutionResults>{execution_results}</ExecutionResults>
Available Variables
——————-
<Variables>{variables_info}</Variables>
——————-
Generate Python code that solves the query based on the current instruction. Your code should build upon previous steps and use the available variables. Use the code API as shown in the example. Enclose your code in <PythonCode></PythonCode> tags. If your code provides a final answer, assign it to a variable named "final_answer".

---

### Prompt G.3: Code Generator in Oneshot Reasoner

You are a helpful assistant specializing in visual reasoning tasks. Your goal is to generate Python code that solves a visual reasoning query using the provided code API and examples.

API Specification
──────────
Use the following code API to guide your solution:
<CodeAPI>{code_api}</CodeAPI>
Task Description
──────────
Review the task description below to understand the problem context:
<TaskDescription>{task_title}{task_desc}</TaskDescription>
Example for Reference
───────────
Here is an example that illustrates the expected format and approach:
<Example>{code_example}</Example>
User Query
───────
This is the query you need to solve:
<Query>{query}</Query>
Extra Context
──────────
<ExtraContext>{extra_context}</ExtraContext>
Code Initialization
────────────
An instance of the "ImagePatch" class is already provided. Use the following initialization code as the starting point:
<ExecutedCode>
image_patch = ImagePatch(image)
</ExecutedCode>
Instruction:
────────
Generate Python code that utilizes the provided API and initialization to solve the query enclosed within the <PythonCode></PythonCode> block. Ensure your solution follows the structure and style of the given example. Ensure the variable "final_answer" is assigned to the result of the query.

## H    DATASET EXAMPLE

### H.1    EXAMPLE FOR VQA

**Example H.1: MATA-SFT-90K Example Input**

<TaskDescription>
Compositional image question answering
This type of question is intended to return a textual answer to the given question.
Please use "final_answer" as the variable name when providing Python code. Make sure "final_answer" is string type.
E.g., For the question "What sport can you use this for?", please provide the name of the sport as your answer in the final step.
E.g., For the question "Is it good weather?", the final answer must be either "yes" or "no".
</TaskDescription>
<Query>Is the tall clock small or large?</Query>
<Instructions></Instructions>
<Feedback>
Detection result: Only one clock has been detected in original_image.
</Feedback>

image_patch = ImagePatch(image)
# Find clock in the image
clock_patches = image_patch.find(["clock"])["clock"]
# Only one clock has been detected
clock_patch = clock_patches[0]

<Variables>
image_patch: ImagePatch(0, 0, 500, 333), patch name: original_image
clock_patches: [ImagePatch(234, 131, 285, 182)]

clock_patch: ImagePatch(234, 131, 285, 182), patch name: clock_1_in_original_image
</Variables>
<StateHistory>
Initial
StepWiseReasoning
StepWiseReasoning
</StateHistory>
<State>StepWiseReasoning</State>
<CurrentStep>3</CurrentStep>
Based on the information above, determine the next state the system should transition to. Choose from the following states:
<StateCandidates>
Final
Specialized
OneShotReasoning
StepWiseReasoning
</StateCandidates>
Return the name wrapped in <NextState> tags.

### Example H.2: MATA-SFT-90K Example Output

<NextState>StepWiseReasoning</NextState>

## H.2 EXAMPLE FOR GROUNDING

### Example H.3: MATA-SFT-90K Example Input

<TaskDescription>
Referring Expression Comprehension
This type of task is to return one image patch in the image that corresponds best to the given query.
The object described by the query must exist in the image, and only have one patch. You need to first detect that kind of object in the image and then identify which one matches the description in the query.
Please use "final_answer" as the target image patch name when providing Python code. Make sure only one ImagePatch in "final_answer".
E.g., query is "left woman with shoes," return one of the detected woman patches in the final step, don't return shoes patch.
E.g., query is "muffins on the table," return one of the muffin patches in the final step, don't return table patch.
E.g., query is "white chaise under window", return one of the chaise patches in the final step, don't return window patch.
</TaskDescription>
<Query>far right</Query>
<Instructions></Instructions>
<Feedback>
Detection result: 5 people have been detected in original_image.
</Feedback>

image_patch = ImagePatch(image)
# Find people in the image
people_patches = image_patch.find(["people"])

<Variables>
image_patch: ImagePatch(0, 0, 640, 427), patch name: original_image
people_patches: {"people": [ImagePatch(374, 0, 584, 377), ImagePatch(0, 7, 153, 353), ImagePatch(200, 47, 361, 408), ImagePatch(517, 0, 640, 382), ImagePatch(113, 174, 195, 353)]}
</Variables>
<StateHistory>
Initial
OneShotReasoning
StepWiseReasoning
</StateHistory>
<State>StepWiseReasoning</State>

<CurrentStep>3</CurrentStep>
Based on the information above, determine the next state the system should transition to. Choose from the following states:
<StateCandidates>
Final
Specialized
OneShotReasoning
StepWiseReasoning
</StateCandidates>
Return the name wrapped in <NextState> tags.

Example H.4: MATA-SFT-90K Example Output

<NextState>StepWiseReasoning</NextState>

## I QUALITATIVE ANALYSIS

We compare MATA with Qwen3-VL (Bai et al., 2025a), ViperGPT (Surís et al., 2023), HYDRA (Ke et al., 2024), and NAVER (Cai et al., 2025). In easy cases (e.g., "find people in red"), the MATA hyper agent transits to a *Specialized* agent that answers directly, and most baselines also succeed. For more complex queries (see Figure 6), stronger compositional reasoning is required; prior methods often hallucinate due to some bottlenecks (e.g., noisy tool outputs, fixed pipelines, no verification).

In **Example 1 (GQA)**, MATA explores with several *Stepwise Reasoner* steps and, after verification failures, hands off the shared memory to the *Oneshot Reasoner* to understand the previous experience, and produce the correct answer. In **Example 2** (zero-shot, generated by GPT-Image), it begins with the *Oneshot Reasoner* to build the initial exploration and save to shared memory, then transitions to the *Stepwise Reasoner*, which first isolates the left table and then counts, again yielding the correct result. These cases illustrate how learned transitions improves robustness.

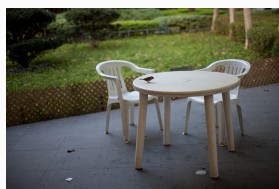

*Example from GQA*

**Query:** *"What's in front of the fence?"* **Ground truth:** *chair*
**Qwen3-VL (4B):** A small bird perched on the table. → ✗ incorrect.
**ViperGPT:** Generates a single program; mis-detects objects "table" → ✗ incorrect.
**HYDRA:** Multistep programming reasoning, but relies on the noisy tools output and unable to fix, producing "There are no objects in front of the fence." → ✗ incorrect.
**NAVER:** Follows fixed Perception→Logic→Answering; cannot produce the text answer → ✗ incorrect.
**MATA (ours):** Initially the hyper agent calls Stepwise Reasoner 3 steps, with detailed exploration and trials, but failed to get the result. Then inherited with the experience of the memory, the hyper agent decides to transit to Oneshot Reasoner, which generates a correct answer based on the previous experience: "white chair" → ✓ correct.

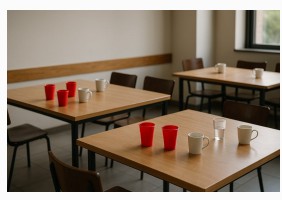

*Example zero-shot*

**Query:** *"How many red cups on the left table?"* **Ground truth:** *3*
**Qwen3-VL (4B):** ... (thinking) There are 4 red cups on the left table. → ✗ incorrect.
**ViperGPT:** Generates a faulty program; no mechanism to fix the program, cannot produce answer → ✗ incorrect.
**HYDRA:** Detects 12 cups and queries color per cup, but never restricts to the left table; answerer then guesses 5 → ✗ incorrect.
**NAVER:** Manually defined Perception→Logic→Answering automaton picks only cup, build the logic that treats "left table" as left of other cups, then collapses to one localization instead of counting; with confidence thresholding and counting to the detections it yields the correct count 3 → ✓ partial correct.
**MATA (ours):** The hyper agent calls Oneshot Reasoner but failed to get a confident answer, then transfers to Stepwise Reasoner invoked 4 times to generate the final answer: 3 → ✓ correct.

Figure 6: **Qualitative comparison.** Previous methods either commit to a single pass (ViperGPT), multi-step within one agent (HYDRA), or follow a fixed automaton (NAVER). **MATA** learns when to *switch agents* and re-enter perception based on shared-memory feedback, yielding robust outcomes on the examples not only from GQA but also the unseen set.