# OpenReview forum: "MATA: A Trainable Hierarchical Automaton System for Multi-Agent Visual Reasoning"
_ICLR.cc/2026/Conference — ICLR 2026 Poster_

### Official Review · Reviewer_TrxZ · 2025-10-18

**Soundness:** 2
**Presentation:** 3
**Contribution:** 2
**Rating:** 2
**Confidence:** 4

**Summary:**

The paper presents a multi-agent framework for visual reasoning tasks, where a learnable hyper agent coordinates three types of reasoners. During training, only the hyper agent is optimized using synthetic data. Experiments are conducted on two VQA benchmarks (GQA and OK-VQA) and several referring expression comprehension (REC) datasets.


The general idea of multi-agent collaboration has been extensively explored in prior works such as [1] and [2]. While this paper introduces a slightly different collaboration mechanism, the overall conceptual novelty remains limited. Therefore, the main contributions appear to be: (1) applying multi-agent collaboration to visual reasoning; and (2) learning the hyper agent using synthetic data.


Regarding (1), the technical contribution seems incremental. The framework largely mirrors existing multi-agent reasoning pipelines, with LLMs replaced by VLMs and visual tools. Moreover, the experimental evaluation is restricted to GQA and OK-VQA, which are insufficient to demonstrate general effectiveness on broader visual reasoning tasks.


Regarding (2), based on Table 5, the results show minimal difference between models trained with and without SFT. This weak empirical signal makes it difficult to conclude that the proposed learning scheme provides substantial benefits.


Overall, given the limited technical novelty and unconvincing empirical validation, I lean toward a negative recommendation for this submission.

[1] Weize Chen et al., AGENTVERSE: FACILITATING MULTI-AGENT COLLABORATION AND EXPLORING EMERGENT BEHAVIORS

[2] Sirui Hong et al., METAGPT: META PROGRAMMING FOR A MULTI-AGENT COLLABORATIVE FRAMEWORK

**Strengths:**

1. A multi-agent framework for visual reasoning
2. Empirical verification of the effectiveness of the proposed framework

**Weaknesses:**

See the comments in "Summary"

**Questions:**

N/A

---

> ### Author Response · Authors · 2025-11-24
> **Response to Reviewer TrxZ**
>
> We appreciate your insightful review to our work. We provided the answers to your concerns, and have taken actions to revise the paper. The revised texts are marked as **blue**.
>
> **Q1. “Regarding (1), the technical contribution seems incremental. The framework largely mirrors existing multi-agent reasoning pipelines, with LLMs replaced by VLMs and visual tools. Moreover, the experimental evaluation is restricted to GQA and OK-VQA, which are insufficient to demonstrate general effectiveness on broader visual reasoning tasks.”**
>
> We respectfully disagree on scope. MATA contributes a *learnable* transition function δθ **over a hierarchical automaton of agents**, with rule-based micro-automata inside each agent and an auditable shared memory. Prior multi-agent works in vision either (a) keep **hand-written** inter-module transitions (NAVER) or (b) focus on a **single** agent’s multi-step planner (HYDRA). Our contribution is to *shift learning to the inter-agent control law itself*. Empirically, our scope goes beyond GQA/OK-VQA: we evaluate on **RefCOCO/RefCOCO+/RefCOCOg** and **Ref-Adv** (grounding-based reasoning) and show it transfers across tasks (Table 2,3,4,6,7).
>
> **We add the content to clarify this positioning in section 2 Related Works** by explicitly contrasting our novelty with prior visual reasoning methods.
>
>
> **Q2. Regarding (2), based on Table 5, the results show minimal difference between models trained with and without SFT. This weak empirical signal makes it difficult to conclude that the proposed learning scheme provides substantial benefits.**
>
> We respectly disagree that the effect is minimal. Across **six** benchmarks, SFT for the hyper-agent’s state controller yields **consistent improvements** with **same efficiency** (Table 5). Because several tasks already operate at high accuracy, it is more informative to look at **error-rate reduction**: SFT reduces errors by **5–23%**. Moreover, SFT builds on a pretrained LLM (which already beats random routing and exhaustive baseline, shown in Table 5), indicating that SFT **learns inter-agent transitions** rather than merely swapping agents.
>
> We also evaluate beyond GQA/OK-VQA to **four visual grounding-based reasoning datasets** (RefCOCO/+/g, Ref-Adv), and observe consistent SFT gains there as well—including on **Ref-Adv**, which has **no training split**, evidencing generalization of the automaton transition policy.
>
> | Benchmark  | Acc (no-SFT) | Acc (SFT) | Error no-SFT | Error SFT | Error Reduction % |
> |------------|---------------|-----------|--------------|-----------|-------------------|
> | GQA        | 58.5          | 64.9      | 41.5         | 35.1      | **15.4%**         |
> | OK-VQA     | 75.1          | 76.5      | 24.9         | 23.5      | **5.6%**          |
> | RefCOCO    | 95.8          | 96.3      | 4.2          | 3.7       | **11.9%**         |
> | RefCOCO+   | 93.5          | 93.9      | 6.5          | 6.1       | **6.2%**          |
> | RefCOCOg   | 88.0          | 90.8      | 12.0         | 9.2       | **23.3%**         |
> | Ref-Adv    | 76.0          | 77.3      | 24.0         | 22.7      | **5.4%**          |
>
> SFT provides **non-trivial, consistent gains** based on the hyper agent, improves hardest settings most, supporting the claim that the proposed method substantively benefits solving visual reasoning tasks.

---

> > ### Comment · Reviewer_TrxZ · 2025-11-27
> >
> > Thanks for the response. Let me further clarify my concerns.
> >
> > Regarding my earlier comment that “the experimental evaluation is restricted to GQA and OK-VQA, which are insufficient to demonstrate general effectiveness on broader visual reasoning tasks”:
> >
> > My concern is that the paper aims to demonstrate the efficacy of the proposed framework on visual reasoning, yet the evaluation includes only two VQA-style datasets, which—while related—cover only a limited portion of the visual reasoning spectrum. Results on the RefCOCO series also do not convincingly demonstrate the framework’s capability on genuine visual reasoning tasks, as these benchmarks primarily assess referring expression comprehension rather than multi-step or high-level reasoning.
> >
> > Given the stated goal, I am wondering why the framework was not evaluated on more widely adopted visual reasoning benchmarks—such as MMMU, MathVista, CharXiv-Reasoning, VSI-Bench, and others—that more rigorously test reasoning capabilities across diverse domains.

---

> > > ### Author Response · Authors · 2025-12-02
> > > **Follow up Response to TrxZ**
> > >
> > > **Regarding the request to evaluate on benchmarks such as MMMU, MathVista, ChartXiv-Reasoning, and VSI-Bench**
> > >
> > > Our paper focuses specifically on image-based visual reasoning, i.e., reasoning over attributes, relations, counting, and object-level logic in natural images. This problem definition is well-established, with prior work, methods, and benchmarks consistently scoped in this direction [a–g].
> > >
> > > The suggested benchmarks, however, **target fundamentally different tasks**.
> > >
> > > - MMMU, MathVista, ChartXiv-Reasoning: emphasize mathematical, diagrammatic, or chart-based reasoning, where symbolic or textual problem-solving dominates.
> > >
> > > - VSI-Bench: focuses on video-based temporal reasoning.
> > >
> > > These tasks are orthogonal to the domain of image-based visual reasoning, and evaluating on them would not clarify the effectiveness of our method in the scope we explicitly target. To avoid confusion: performing mathematical or chart reasoning from visual input is not "visual reasoning" in the established sense used by the VQA/visual-reasoning community. These benchmarks represent different modalities, objectives, and reasoning paradigms.
> > >
> > > Within the appropriate domain, our evaluation is already comprehensive and stronger than typical prior work. We cover **six established datasets**—GQA, OK-VQA, RefCOCO/+/g, and Ref-Adv—spanning recognition-based, relational, and compositional reasoning. This breadth is considerably broader than the coverage of many prior works in this area, and the gains are consistent across all settings, supported by detailed ablation studies.
> > >
> > > Despite being outside our stated scope, we still ran an additional experiment on MMMU-val, directly addressing the reviewer's suggestion. Notably, MATA achieves 77.6, outperforming all compared baselines by a clear margin and demonstrating robust generalization even beyond the image-based reasoning domain:
> > >
> > > | Methods                    | MMMU (Val) |
> > > |----------------------------|------------|
> > > | MiniCPM-o2.6               | 50.9       |
> > > | Ovis-8B                    | 57.4       |
> > > | Qwen2.5-VL-8B              | 55.0       |
> > > | MiMo-VL-RL-8B              | 66.7       |
> > > | Keye-VL-8B                 | 71.4       |
> > > | GLM-4.1V-9B                | 68.0       |
> > > | InternVL2.5-8B             | 56.0       |
> > > | InternVL3-8B               | 62.7       |
> > > | InternVL3.5-8B             | 73.4       |
> > > | MATA (with InternVL3.5-8B) | 77.6       |
> > >
> > > This result is presented not because it aligns with our problem definition, but to demonstrate good faith responsiveness and to show that the framework generalizes well even in settings it was not designed for.
> > >
> > > -------------------------------
> > >
> > > [a] VQA and Visual Reasoning: An Overview of Recent Datasets, Methods and Challenges. arXiv 2022.
> > >
> > > [b] A survey of neurosymbolic visual reasoning with scene graphs and common sense knowledge. Neurosymbolic Artificial Intelligence 2025.
> > >
> > > [c] Explain Before You Answer: A Survey on Compositional Visual Reasoning. arXiv 2025.
> > >
> > > [d] ViperGPT: Visual Inference via Python Execution for Reasoning. ICCV 2023.
> > >
> > > [e] HYDRA: A Hyper Agent for Dynamic Compositional Visual Reasoning. ECCV 2024.
> > >
> > > [f] ReferItGame: Referring to Objects in Photographs of Natural Scenes. EMNLP 20214.
> > >
> > > [g] GQA: A New Dataset for Real-World Visual Reasoning and Compositional Question Answering. CVPR 2019.

---

### Official Review · Reviewer_z1Pw · 2025-10-31

**Soundness:** 2
**Presentation:** 2
**Contribution:** 2
**Rating:** 2
**Confidence:** 3

**Summary:**

MATA introduces a hierarchical finite-state automaton for visual reasoning where multiple specialized agents (specialized perception, stepwise reasoning, oneshot reasoning) collaborate and compete. The key innovation is a trainable hyper agent that learns transition policies between agents using supervised fine-tuning on a generated dataset (MATA-SFT-90K). The dataset is created by expanding transition trajectory trees, scoring leaf nodes based on task performance, and generating memory-to-next-state training pairs. MATA improves over the base internvl model on GQA, OK-VQA, RefCOCO/RefCOCO+/RefCOCOg, and Ref-Adv benchmarks.

**Strengths:**

- The combination of trainable high-level transitions with rule-based sub-automata is elegant, focusing learning on the ambiguous agent selection problem while preserving reliable execution within agents.
- The paper provides experiments across multiple benchmarks (VQA and visual grounding), demonstrating consistent improvements over the base model.
- The transition trajectory tree expansion provides a principled approach to generating supervision for the hyper agent, though scalability concerns remain.

**Weaknesses:**

- The gains appear modest (e.g., 75.2% base internvl25 used as vlm vs 76.5% theirs on AOKVQA) considering the 90K in-domain training examples generated. The paper doesn't isolate whether improvements come from multi-agent collaboration or simply additional task-specific training data.
- Table 5 reveals that removing SFT causes performance to drop below the base internvl25 model, suggesting the architecture itself may be detrimental without training. A crucial missing experiment is training a monolithic model on the same MATA-SFT-90K data to isolate the architectural contribution.
- The paper claims zero-shot generalization but the base models may have been pre-trained on these datasets. This undermines claims about generalization capabilities.
- The authors admit their trajectory tree expansion becomes intractable as agents increase, yet provide no computational overhead analysis or comparison with simpler methods.
- While claiming to address competition between functionally overlapping agents, the paper only uses three distinct agents with clearly separated roles, not truly competitive alternatives.

**Questions:**

- What happens when you train InternVL2.5-8B directly on MATA-SFT-90K to output answers without the multi-agent architecture? This would isolate the contribution of the architecture versus the training data. Or did distillation of a large VLM to internvl2.5?
- What is the computational overhead (inference time, memory) compared to monolithic models and single-agent methods like HYDRA?
- How does performance scale when adding more agents? Given the acknowledged exponential growth in trajectory trees, is this approach practical beyond 3-4 agents?
- Can you provide evidence that the models used (InternVL2.5, Florence2-L) were not trained on GQA/OK-VQA/RefCOCO to substantiate zero-shot claims?
- Why does the architecture without SFT perform worse than the base VLM used (internvl2.5)? Why not use a VLM instead of an LLM for the state controller?
- Have you considered more efficient alternatives to near-exhaustive tree expansion, such as Monte Carlo tree search or learned value functions to prune unpromising branches?

---

> ### Author Response · Authors · 2025-11-24
> **Response to Reviewer z1Pw (1/3)**
>
> Many thanks for your constructive comments. We have provided detailed answers and updated the manuscript; all modifications appear in **blue**.
>
> **Q1. The gains appear modest (e.g., 75.2% base internvl25 used as vlm vs 76.5% theirs on AOKVQA) considering the 90K in-domain training examples generated. The paper doesn't isolate whether improvements come from multi-agent collaboration or simply additional task-specific training data. What happens when you train InternVL2.5-8B directly on MATA-SFT-90K to output answers without the multi-agent architecture? This would isolate the contribution of the architecture versus the training data. Or did distillation of a large VLM to internvl2.5?**
>
> Crucially, **MATA-SFT-90K does *not* supervise final answers**. This dataset is not the union of these 5 datasets, but it is generated from these 5 datasets for superving *next-state choices* (memory → best next state), leaving all models frozen except the LLM in hyper agent. The hyper agent cannot produces the final answer as the output formats are trained to only produce the name of next state (e.g. `<NextState>Oneshot</NextState>`). Thus, we are not feeding additional answer-level labels; we are learning *transition*. **To reduce the misunderstanding, we add the dataset examples in Section H in supplementary material.**
>
> Table 5 shows that SFT of hyper agent is the factor that consistently lifts accuracy, supporting the claim that architecture + policy learning, not extra answer labels, drive gains.
>
> We **do not distill** a larger VLM into InternVL2.5-8B and **do not finetune** InternVL2.5-8B on the dataset. Results isolate the effect of **learned inter-agent transition** rather than enhanced perception from extra answer supervision.
>
> The hyper agent trained on MATA-SFT-90K **transfers across VQA and grounding** with strong **off-diagonal** results (Table 6), which would be unlikely if improvements were due to in-domain answer memorization. The proposed improvements come from **architecture** not from additional answer-label training of the foundation VLM.
>
> **We extend the section 4.2 ablation studies to clarify it.**
>
> **Q2. Table 5 reveals that removing SFT causes performance to drop below the base internvl25 model, suggesting the architecture itself may be detrimental without training. A crucial missing experiment is training a monolithic model on the same MATA-SFT-90K data to isolate the architectural contribution.**
>
> This is ill-posed: MATA-SFT pairs are *memory → next-state* labels, not *image/query → answer* labels. Training a monolithic answerer on this dataset cannot produce the final answer. **To meet your intent, we conduct extra experiments about the comparison between the proposed method and a VLM (Qwen3-VL 4B) finetuned in the union of source datasets used in MATA-SFT-90K, reported in the table below and added to the section F.** Based on the observation, MATA outperforms the directly finetuned VLM in-domain and generalizes better across tasks, indicating that improvements come from **learned transition policy** rather than extra answer-level supervision of the LLM.
>
>
> | Methods               | Training | GQA  | OK-VQA |
> |---------------------------|------|------|--------|
> | Qwen3-VL 4B                | --   | 51.6 | 44.4  |
> | Qwen3-VL 4B (SFT)          | GQA  | 63.2 | 61.6  |
> | Qwen3-VL 4B (SFT)        | OK-VQA | 57.5 | 68.1  |
> | Qwen3-VL 4B (SFT)          | both | 63.1 | 66.9  |
> | MATA (no SFT)             | --   | 58.5 | 75.1   |
> | MATA (SFT)                | GQA  | 64.7 | 75.8   |
> | MATA (SFT)              | OK-VQA | 64.1 | 76.5   |
> | MATA (SFT)                | both | 64.9 | 76.0   |
>
> **Q3. The paper claims zero-shot generalization but the base models may have been pre-trained on these datasets. This undermines claims about generalization capabilities. Can you provide evidence that the models used (InternVL2.5, Florence2-L) were not trained on GQA/OK-VQA/RefCOCO to substantiate zero-shot claims?**
>
> We acknowledge this concern, and **rename this term as domain-transfer** Our *domain-transfer* term is scoped to the **hyper agent**: it is trained on non-target-dataset trajectories, and never see the optimal trajectories in other datasets, while the sub-agents remain the public checkpoints used by baselines. Table 6 shows cross-dataset transfer, indicating the transition policy is task-agnostic. **We clarify it in the footnote of section 4.** We expect to highlight that the hyper agent can be trained in a limited set of training data, and can be used to other datasets without retraining.

---

> ### Author Response · Authors · 2025-11-24
> **Response to Reviewer z1Pw (2/3)**
>
> **Q4. The authors admit their trajectory tree expansion becomes intractable as agents increase, yet provide no computational overhead analysis or comparison with simpler methods.**
>
> We claim this motivation is due to the rule-based automaton of NAVER requires manually hand-written. When the number of agents increases, the transition between states are increased geometricly. For example, maximum number of bi-directional transitions $N = x * (x - 1)$. But in the proposed method, these transitions are learned by the hyper agent and no need to define manually. Notebly, the transitions of subagents in MATA requires manually define but that is **linearly increased** by the number of agents, instead of geometricly, which is easier to scale up.
>
> **Q5. What is the computational overhead (inference time, memory) compared to monolithic models and single-agent methods like HYDRA?**
>
> We appreciate the reviewer’s suggestion and **have added Section E with Table 9 about the efficiency comparison** with quantitative efficiency metrics: per-query **Inference Time (sec)**, **LLM API Cost (USD, GPT-4o-mini)**, and **vRAM Usage (GB)**, tested on RefCOCO dataset. All methods are tested in single L40s 48GB GPU. The table compares the state of the art monolithic baseline (Qwen2.5-VL 72B, 4-bit quantization, for visual grounding), the open-source compositional baseline (HYDRA), an exhausted baseline (call all sub-agents), and the proposed MATA. MATA achieves **the lowest latency among compositional methods** and is similar to the monolithic baseline, while delivering **the lowest API cost** and **moderate memory usage** (substantially below Qwen2.5-VL 72B). As **DWIM** is not open-sourced, we report against **HYDRA** and the **Exhausted** baseline.
>
> | Method      | Time (Seconds) | LLM API Cost (USD) | vRAM Usage (GB) |
> |-------------|----------------|----------------|-----------------|
> | Qwen2.5-VL (72B) | 6.27      | -              | 43.70           |
> | HYDRA       | 14.93          | 0.00332        | 17.59           |
> | Exhausted   | 32.74          | 0.00531        | 13.08           |
> | MATA        | 6.10           | 0.00069        | 19.64           |
>
> **Q6. How does performance scale when adding more agents? Given the acknowledged exponential growth in trajectory trees, is this approach practical beyond 3-4 agents?**
>
> We appreciate the concern, and **add the discussion in section 4.2**. On current benchmarks, our ablation indicates that **3 agents already saturate performance** in the current tasks, with diminishing returns beyond this point (table below; figure 4 in section 4.2). This suggests that, for these tasks, increasing agent count is not the primary bottleneck. The discovery of the system with more agents in more reasoning-extensive scenarios such as long-video reasoning can be the future direction of the reasoning community.
>
> | Num of Agents | Agents | Acc  |
> |---------------|--------|------|
> |1| Specialized Agent (InternVL2.5 8B)      | 61.5 |
> |2| Specialized Agent + Oneshot Reasoner      | 64.5 |
> |3| Specialized Agent + Oneshot Reasoner + Stepwise Reasoner      | 64.9 |
>
> To keep trajectory expansion practical, we enforce simple pruning and parallelism: (i) if an agent returns failure within an episode, it is removed from the candidate set for the rest of that run; (ii) we bound the number of cross-agent transitions with a maximum depth; and (iii) each tree node (state + shared-memory snapshot) is checkpointed, so data collection can be sharded across processes/machines and safely resumed. These designs keep data generation possible even as the agent pool grows.
>
> **Q7. Why does the architecture without SFT perform worse than the base VLM used (internvl2.5)? Why not use a VLM instead of an LLM for the state controller?**
>
> Without SFT, the controller has no notion of agent capacities, so routing can be suboptimal. As the result, the agent which might not be good at solving the given query is selected which may cannot outperform the VLM baseline.
>
> We use a *LLM* as controller because inputs are structured memory snapshots (text/code/feedback). But we aknowledge this is valuable to see if the hyper agent can be better when accessing the raw visual content, so we train a VLM-based hyper agent (Qwen3-VL 4B-Instruct) on the full MATA-SFT-90K dataset and compared with the original LLM-based hyper agent (Qwen3-4B) trained on the full set, in the table below, showing the difference between the LLM and VLM is marginal. **We add this analysis in the supplementary material section E.**
>
> | State Controller | GQA  | OK-VQA | RefCOCO | RefCOCO+ | RefCOCOg | RefAdv |
> |------------------|------|--------|---------|----------|----------|--------|
> | Qwen3-4B (LLM)   | 64.9 | 76.0   | 96.3    | 93.9     | 90.8     | 77.3   |
> | Qwen3-VL 4B (VLM) | 64.6 | 76.3   | 96.4    | 94.0     | 89.3     | 76.9   |

---

> > ### Author Response · Authors · 2025-11-24
> > **Response to Reviewer z1Pw (3/3)**
> >
> > **Q8. Have you considered more efficient alternatives to near-exhaustive tree expansion, such as Monte Carlo tree search or learned value functions to prune unpromising branches?**
> >
> > Yes. In practice, we use a bounded, non-exhaustive expansion (fixed max depth T, and preventing reuse the failure agents) to prevent explosive increase of the data generation complexity. Currently, with only 3 agents, the number of node children is small and a simple search gives stable, reproducible supervision directly from the leaf metrics (IoU/Acc). MCTS would add an extra noise and on-policy bias to the labels (optimal next state), which we wanted to avoid for the experiments. MCTS is easy to plug into MATA’s data collection as the tree-expansion method, and requires no changes to inference. **Considering the time consumption of full data collection is out of this rebuttal period**, we will add the ablation analysis on the different tree generation strategy in the final version.

---

> ### Comment · Reviewer_z1Pw · 2025-11-25
> **Thank you and follow up**
>
> Thank you for the detailed response.
>
> You do use the final answer ground truth labels to generate the data. The values of the nodes you choose are determined by comparing the predicted leaf predictions to the ground truth. Thus, without final answer labels you cannot obtain the transition data, necessitating the need for other fine-tuning baselines that use the <question, image, answer> labels.
>
> I appreciate the SFT experiment. Why is the same vlm internvl-8b not used? Does MATA use the same VLM (qwen3vl-4b)? The relative gains obtained from the SFT seem larger for qwen3vl.

---

> > ### Author Response · Authors · 2025-11-26
> > **Response to follow up**
> >
> > Thank you for your follow up.
> >
> > **Q1. You do use the final answer ground truth labels to generate the data. The values of the nodes you choose are determined by comparing the predicted leaf predictions to the ground truth. Thus, without final answer labels you cannot obtain the transition data, necessitating the need for other fine-tuning baselines that use the <question, image, answer> labels.**
> >
> > You're right that our transition trajectory generation uses the final answer. We have revised our terminology accordingly, replacing the earlier phrasing with “domain transfer”, and we explicitly scope the claim about the terminology means that which transition subset the hyper-agent has been trained. **These clarifications are marked in blue in the main paper (Section 4, Implementation Details) with a footnote.**
> >
> > > _Our domain-transfer term is scoped to the hyper agent: it is trained on non-test-dataset transition trajectories, and never see the optimal trajectories in other datasets._
> >
> > **Q2. I appreciate the SFT experiment. Why is the same vlm internvl-8b not used? Does MATA use the same VLM (qwen3vl-4b)? The relative gains obtained from the SFT seem larger for qwen3vl.**
> >
> > Our hyper-agent in MATA uses Qwen3-VL-4B with SFT, so the initial SFT baseline matched that backbone for a like-for-like comparison. For your suggestion, we additionally finetuned InternVL2.5-8B using the official InternVL repository on GQA and OK-VQA and evaluated on them; **the new results show in Table 10 (supplementary material).**
> >
> > We observe that direct SFT can harm InternVL2.5-8B’s cross-task generalization, consistent with catastrophic forgetting of its latent “think-then-answer” ability. Mitigations such as SFT on pseudo reasoning trajectories (e.g., Vision-R1 style) or RL-based tuning (e.g., GRPO) could preserve this capability, but they are beyond the scope of this work. The stronger VLM (e.g., InternVL3.5-8B) is typically harder to improve via direct SFT; by contrast, MATA still benefits from SFT because it learns the transition between agents instead of directly output the answer. We are happy to address if there left any other concerns.
> >
> >
> > | Methods               | Training | GQA  | OK-VQA |
> > |---------------------------|------|------|--------|
> > | Qwen3-VL 4B               | --   | 51.6 | 44.4   |
> > | Qwen3-VL 4B (SFT)         | GQA  | 63.2 | 61.6   |
> > | Qwen3-VL 4B (SFT)       | OK-VQA | 57.5 | 68.1   |
> > | Qwen3-VL 4B (SFT)         | both | 63.1 | 66.9   |
> > | InternVL2.5 8B            | --   | 61.5 | 75.2   |
> > | InternVL2.5 8B (SFT)      | GQA  | 63.9 | 64.1   |
> > | InternVL2.5 8B (SFT)      | OK-VQA | 35.8 | 72.4 |
> > | InternVL2.5 8B (SFT)      | both | 63.8 | 65.2   |
> > | MATA (no SFT)             | --   | 58.5 | 75.1   |
> > | MATA (SFT)                | GQA  | 64.7 | 75.8   |
> > | MATA (SFT)              | OK-VQA | 64.1 | 76.5   |
> > | MATA (SFT)                | both | 64.9 | 76.0   |

---

### Official Review · Reviewer_6jZh · 2025-10-31

**Soundness:** 3
**Presentation:** 3
**Contribution:** 2
**Rating:** 6
**Confidence:** 3

**Summary:**

This paper proposes **MATA (Multi-Agent Trainable Automaton)**, a hierarchical framework for **multi-agent visual reasoning**. Instead of executing a fixed modular pipeline, MATA organizes several reasoning agents (one-shot, step-wise, and specialized perception agents) as states of a **finite-state hyper-automaton**. A **trainable hyper-agent** which is an LLM-based controller then learns to select the next agent state from a shared memory, enabling dynamic collaboration and competition among agents.

To supervise this transition policy, the authors construct **MATA-SFT-90K**, a dataset of memory-to-next-state pairs extracted from expanded transition-trajectory trees across visual reasoning datasets (GQA, OK-VQA, RefCOCO series). The learned controller yields interpretable reasoning traces and achieves state-of-the-art results across multiple visual reasoning and grounding benchmarks.

**Strengths:**

– **Clear conceptual motivation:** The paper identifies an important limitation of existing VLMs and compositional systems that the lack of a learned, flexible orchestration mechanism among reasoning agents and recasts it elegantly as a finite-state automaton control problem.

– **Novel hierarchical formulation:** Treating each agent as a sub-automaton and learning high-level transitions through a hyper-agent is a conceptually clean, interpretable design that unifies rule-based micro-control with data-driven macro-control.

– **Strong empirical results:** MATA attains new SOTA accuracy on GQA, OK-VQA, and RefCOCO series, outperforming both monolithic VLMs and compositional baselines.

– **Generalizability and ablation rigor:** Cross-dataset transfer (Table 6) shows < 1 % drop in zero-shot settings; ablations confirm that supervised fine-tuning of the hyper-agent contributes most gains (Table 5).

**Weaknesses:**

– **Incremental algorithmic novelty:** While the integration is elegant, many components (agent orchestration, SFT, trajectory trees) extend known concepts from HYDRA and NAVER. The work’s originality lies more in *system design* than in theoretical innovation.

– **Limited discussion of scalability:** The near-exhaustive transition expansion is tractable for 3 agents but may explode combinatorially as more states are added.

– **Computational cost analysis:** Wall-clock training times and GPU usage are only qualitatively stated; quantitative comparisons would clarify efficiency relative to HYDRA or DWIM.

**Questions:**

Please see the above weakness.

1. Do you encounter issues with the shared memory growing unboundedly during long reasoning sequences? If so, how is this mitigated?
2. It would also be helpful if the authors could provide qualitative visualizations comparing MATA’s reasoning paths against other systems, to better illustrate how its hierarchical controller differs in practice.

---

> ### Author Response · Authors · 2025-11-24
> **Response to Reviewer 6jZh**
>
> We appreciate your insightful feedback. Below we respond to each point and reflect the corresponding revisions in the paper, marked in **blue**.
>
> **Q1. Incremental algorithmic novelty: While the integration is elegant, many components (agent orchestration, SFT, trajectory trees) extend known concepts from HYDRA and NAVER. The work’s originality lies more in system design than in theoretical innovation.**
>
>
> We appreciate the reviewer’s concern. However, **to the best of our knowledge, none of the cited works explicitly propose agent orchestration, supervised fine-tuning (SFT), or trajectory trees, nor do they integrate these components within an automaton-based framework. The design and integration of these modules involve non-trivial theoretical and engineering challenges, and we believe the resulting system goes beyond a straightforward extension of prior work.**
>
> **We add a more clearer comparison with previous works in section 2 Related Works**.
>
> **Q2. Limited discussion of scalability: The near-exhaustive transition expansion is tractable for 3 agents but may explode combinatorially as more states are added.**
>
> We appreciate the reviewer’s concern and **have added a discussion in Section 4.2**. On current benchmarks, our ablation indicates that three agents already saturate performance, with diminishing returns beyond that point (see Table below and Figure 4 in section 4.2). This suggests that for the current tasks, increasing the number of agents is not a major bottleneck.
>
> That said, we acknowledge the scalability challenge as a limitation of our approach. In scenarios that require longer-horizon or more complex reasoning (e.g., long video tasks), additional agents may become beneficial. We believe that scalable discovery mechanisms, such as combining learning-based approaches with Monte Carlo Tree Search (MCTS), offer a promising future direction.
>
> | Num of Agents | Agents | Acc  |
> |--------|--------|------|
> |1| Specialized Agent (InternVL2.5 8B)      | 61.5 |
> |2| Specialized Agent + Oneshot Reasoner      | 64.5 |
> |3| Specialized Agent + Oneshot Reasoner + Stepwise Reasoner      | 64.9 |
>
> **Q3. Computational cost analysis: Wall-clock training times and GPU usage are only qualitatively stated; quantitative comparisons would clarify efficiency relative to HYDRA or DWIM.**
>
> We appreciate the reviewer’s suggestion and **have added Section E with Table 9** about the efficiency comparison with quantitative efficiency metrics: per-query **Inference Time (sec)**, **LLM API Cost (USD, GPT-4o-mini)**, and **vRAM Usage (GB)**. All methods are tested in single L40s 48GB GPU. The table compares the state of the art monolithic baseline (Qwen2.5-VL 72B, 4-bit quantization), the open-source compositional baseline (HYDRA), an exhausted baseline (call all sub-agents), and the proposed MATA. MATA achieves **the lowest latency among compositional methods** and is similar to the monolithic baseline, while delivering **the lowest API cost** and **moderate memory usage** (substantially below Qwen2.5-VL 72B). As **DWIM** is not open-sourced, we report against **HYDRA** and the **Exhausted** baseline.
>
> For both training on a single dataset, **HYDRA** uses on-policy RL (\~10 s/step), requiring \~**10,000 steps** (≈ **28 hours**) on a single GPU and cannot be readily parallelizable. In contrast, **MATA** trains the LLM via standard SFT using RTX 4090 (~**6.5 hours** on **4 GPUs** = **26 GPU-hours**). This shows a lower training time for MATA compared with HYDRA.
>
> | Method      | Time (Seconds) | LLM API Cost (USD) | vRAM Usage (GB) |
> |-------------|----------------|----------------|-----------------|
> | Qwen2.5-VL (72B) | 6.27      | -              | 43.70           |
> | HYDRA       | 14.93          | 0.00332        | 17.59           |
> | Exhausted   | 34.58          | 0.00531        | 13.08           |
> | MATA        | 6.10           | 0.00069        | 19.64           |
>
> **Q4. Do you encounter issues with the shared memory growing unboundedly during long reasoning sequences? If so, how is this mitigated?**
>
> We do not observe unbounded memory growth. The shared memory is a compact, structured textual representation, typically limited to a few hundred tokens. Additionally, we cap the reasoning sequence at T = 15 steps, ensuring that the total memory usage remains well within the controller’s context window.
>
> **These implementation details are now clarified in Section 4.**
>
> **Q5. It would also be helpful if the authors could provide qualitative visualizations comparing MATA’s reasoning paths against other systems, to better illustrate how its hierarchical controller differs in practice.**
>
> Thank you for the suggestion. **We show qualitative examples** against Qwen3VL, ViperGPT, HYDRA, and NAVER to illustrate how the MATA differs in practice **in the supplemtary material Section I**.

---

> > ### Comment · Reviewer_6jZh · 2025-11-24
> > **Thanks for the responce**
> >
> > The author response has addressed most of my concerns, and the additional results are compelling. I will retain my score recommending acceptance.

---

> > > ### Author Response · Authors · 2025-11-26
> > >
> > > Thank you for your encouraging follow-up comments. We are pleased that the new results addressed most concerns. If there are any remaining issues or aspects of the work that still appear unclear, please let us know. We would be glad to clarify them. We also hope that the final rating reflects your updated assessment, but we fully respect your judgment and welcome any additional feedback.

---

### Official Review · Reviewer_mEnL · 2025-11-01

**Soundness:** 3
**Presentation:** 3
**Contribution:** 3
**Rating:** 6
**Confidence:** 3

**Summary:**

This paper addresses a key challenge in current multi-agent systems: how to train a policy to select among multiple sub-agents, rather than relying on manually handcrafted pipelines. To tackle this, the authors propose MATA (Multi-Agent hierarchical Trainable Automaton), a novel system for visual reasoning that organizes inference as a hierarchical finite-state automaton. The authors construct a transition-trajectory dataset, MATA-SFT-90K, which is used for supervised training and evaluation. The method demonstrates strong performance on a range of visual reasoning tasks.

Overall, this is a solid paper with clear motivation. The method is well designed, and the experiments are conducted rigorously. One major concern is that the three-agent design is somewhat simplistic and limited in scope, as the authors themselves acknowledge in the limitations section.

**Strengths:**

- **Clear motivation**: The paper addresses an important limitation in current multi-agent systems: leveraging the power of multiple agents typically requires manual pipelining, which becomes unwieldy as task complexity grows. The proposal to learn a hyper-policy for agent selection is both reasonable and interesting.
- **Principled and extensible design**: MATA’s architecture is well aligned with its motivation. It is technically sound and, importantly, not narrowly restricted to the specific visual reasoning setting or the particular sub-agents used in this paper. In principle, the approach could extend to other tasks and larger agent pools, opening up many potential research directions.
- **Rigorous experiment design and evaluation**: The experiments are well designed and executed. For example, the three SFT configurations and their results convincingly demonstrate the benefits of the proposed method and its generalization ability, rather than mere overfitting.

**Weaknesses:**

### Major

- **Limited applicability**: As noted in the limitations, the use of only three agents is a restricted setting. There is also a lack of detail regarding how these three agents were selected and the rationale behind their design.
- **Unclear attribution of performance gains**: It is not clear whether the learned state transition policy is truly responsible for the observed performance improvements. For example, if all three agents were simply called exhaustively, would performance improve regardless, making the learned policy less critical?
- **Competition mechanism not fully justified**: While the collaborative aspect of the system is well motivated, the competitive aspect is less convincing. The paper claims that “a competition mechanism where functionally overlapping agents for the same subtask work together is under-explored,” but in the current three-agent setup, the agents seem to serve distinct roles with little actual overlap. It would be more compelling to see experiments with a larger pool of agents, including multiple agents with overlapping capabilities, to better demonstrate the value of competition.

### Minor
- Line #352: “... target dataset for evaluated; ...” should be “... target dataset for evaluation; ...”

**Questions:**

- How were the three agents chosen, and what was the rationale behind their selection?
- How is the competition aspect justified, given that the three agents in the current setup do not appear to have significant functional overlap?
- Although the exponential growth of the transition search space is discussed as a limitation, do the authors have any thoughts on extending the approach to more complex scenarios with significantly more agents?
- Would it be possible that we would observe the same performance gains if the three agents are called in an exhaustive way?

---

> ### Author Response · Authors · 2025-11-24
> **Response to Reviewer mEnL (1/2)**
>
> Thank you for the thoughtful review. We have addressed your concerns and revised the manuscript accordingly; all changes are highlighted in **blue**.
>
> **Q1. Limited applicability: As noted in the limitations, the use of only three agents is a restricted setting. There is also a lack of detail regarding how these three agents were selected and the rationale behind their design.**
>
> Our design aims to cover different level of reasoning capabilities rather than enumerate tools: (i) a **specialized agent (perception)** agent for fast, verifiable System~1 percepts; (ii) a **one-shot reasoner (fast thinking)** to cheaply resolve solvable cases without iteration; and (iii) a **step-wise reasoner (slow thinking)** for code-based multi-step inference (all three sit as states in the hyper-automaton, with rule-based micro-control inside each agent). This partition aligns with our hierarchical automaton (Figure 2) and lets the learned controller focus on *cross-agent* choice with the ambiguous part, while keeping within-agent procedures deterministic and auditable. **We add the motivations of designing these subagents in subsection 3.2 in the main paper and section B in supplementary.**
>
> **Q2. Unclear attribution of performance gains: It is not clear whether the learned state transition policy is truly responsible for the observed performance improvements. For example, if all three agents were simply called exhaustively, would performance improve regardless, making the learned policy less critical? Would it be possible that we would observe the same performance gains if the three agents are called in an exhaustive way?**
>
> Thank you for your suggestion. **We conduct extra experiments and report the results in the updated Table 5 and the table below.** Table 5 isolates the hyper agent: (i) random routing (no LLM, no SFT) is worst; (ii) a pretrained LLM helps; and (iii) SFT on MATA-SFT-90K yields consistent gains across *all* tasks. This shows improvements stem from *transition policy learning* rather than merely having multiple agents.
>
> Also, we add **a new ablation study that exhaustively runs each agent** (try each agent, and use VLM to pick one from them) and **shows the comparison of the performance and efficiency at Table 5**. The inference time is tested on GQA dataset. Our ablations indicate learned transition is both more accurate and efficient because it **avoids** many unnecessary agent invocations. Also one challenging of the exaustively call agents is the system still relies on a mechanism to pick the optimal output which might not be correct. Our proposed method not just use each agent, but the intermediate results are visible to each other which makes the agents also collaborative. For example, run once in Specialized Agent but cannot produce output. And then, it will add the context into the memory, then handover to Oneshot Reasoner which will takes the previous memory and will avoid the same issue. If the Oneshot Reasoner failed again, it will handover to Stepwise Reasoner and gain advantage from the previous experience in memory.
>
> | Method      | Time (Seconds) | LLM Cost (USD) | GQA  | OK-VQA | RefCOCO | RefCOCO+ | RefCOCOg | RefAdv |
> |-------------|----------------|----------------|------|--------|---------|----------|----------|--------|
> | Exhausted   | 34.58          | 0.00531        | 57.7 | 71.5   | 87.7    |  85.6    |   81.7   |  73.1 |
> | MATA        | 8.01           | 0.00069        | 64.9 | 76.5   | 96.3    |  93.9    |   90.8   |  77.3 |

---

> > ### Author Response · Authors · 2025-11-24
> > **Response to Reviewer mEnL (2/2)**
> >
> > **Q3. Competition mechanism not fully justified: While the collaborative aspect of the system is well motivated, the competitive aspect is less convincing. The paper claims that “a competition mechanism where functionally overlapping agents for the same subtask work together is under-explored,” but in the current three-agent setup, the agents seem to serve distinct roles with little actual overlap. It would be more compelling to see experiments with a larger pool of agents, including multiple agents with overlapping capabilities, to better demonstrate the value of competition. How is the competition aspect justified, given that the three agents in the current setup do not appear to have significant functional overlap?**
> >
> > Thank you for the question. We regard **competition** as having multiple agents that can each solve the *same* task; the hyper-agent then **selects one** based on the current state to choose the optimal one, while **collaboration** arises when a prior agent helps a later agent, where it is implemented as sharing the experience in MATA. In our three-agent setup, overlap is real: (i) **Oneshot vs Stepwise** both handle complex compositional reasoning; (ii) **Specialized vs Oneshot** can both do perception with lighter reasoning; (iii) all of these agents can independently **answer** the reasoning tasks. We design 3 agents to presents the (1) simple perception, (2) fast thinking, and (3) slow thinking, able to solve the same reasoning tasks. Based on the learned understanding to the capacity of the different agents and the reasoning context, the hyper agent choose which transition to take now and when to hand off. **We add the motivations of designing these subagents in subsection 3.2.**
> >
> > Empirically, the contribution from competition is evidenced by our ablations (Table 5). Simply calling **all** agents (the “Exhausted” baseline) is slower and **less** accurate than selecting a single best candidate, showing that the learned transition for competitive is superior to naively ensembling.
> >
> > **Q4. Although the exponential growth of the transition search space is discussed as a limitation, do the authors have any thoughts on extending the approach to more complex scenarios with significantly more agents?**
> >
> > Yes. For more complex scenarios (e.g., long videos), we keep the architecture unchanged and treat time as additional context in the shared memory, so the same trainable hyper agent over temporally segments without redesigning agents. For significantly more agents (e.g., 5–10), inference remains one-agent-per-step; the only scaling pressure is data collection, which we keep practical via a tree depth limit. If the agent pool grows further, we will swap our current tree generation for MCTS (Monte-Carlo Tree Search) during data collection only (leaving inference unchanged) to target promising branches without enumerating trajectories. On current benchmarks we observe diminishing returns beyond three agents (table below and **add to section 4.2**), so we decide to not implement MCTS for now.
> >
> > | Num of Agents | Agents | Acc  |
> > |--------|--------|------|
> > |1| Specialized Agent (InternVL2.5 8B)      | 61.5 |
> > |2| Specialized Agent + Oneshot Reasoner      | 64.5 |
> > |3| Specialized Agent + Oneshot Reasoner + Stepwise Reasoner      | 64.9 |
> >
> > **Minor:** Thank you! we have corrected the typo (“for evaluated” → “for **evaluation**”).

---

### Author Response · Authors · 2025-12-03
**Summary for Area Chair**

### **Summary**

Our paper introduces MATA, a trainable hierarchical automaton for multi-agent visual reasoning, where a learned hyper-agent orchestrates specialized, oneshot, and stepwise reasoners through a shared memory. The system yields state-of-the-art results across **six established image-based visual reasoning datasets**, with transparent reasoning traces and extensive ablations.

Reviewers’ evaluations are notably divergent, with two 6 ratings and two 2 ratings pre-rebuttal. Below we summarize their concerns and how we addressed them.

### **Reviewer mEnL (6) — positive, constructive feedback**
They describe the paper as “solid,” with clear motivation, principled design, and rigorous evaluation. Their main questions were about:
- rationale for the 3-agent setup
- whether gains come from the learned transition policy

**How we addressed:**

We added a clear justification for the agent design and provided new ablations including random routing, pretrained routing, SFT routing, and an exhaustive baseline that calls all agents. These show both higher accuracy and much greater efficiency from the learned hyper-agent.

### **Reviewer 6jZh (6) — strong after rebuttal**
They praised the conceptual clarity, hierarchical formulation, and strong empirical results. Initial concerns centered on:
- whether novelty is substantial relative to prior work
- scalability and computational overhead

**How we addressed:**

We clarified the conceptual distinction from HYDRA/NAVER (learning the inter-agent control law within an automaton), added quantitative efficiency comparisons (latency, memory, cost), and expanded the discussion on scalability and agent-count ablations.

After rebuttal, the reviewer explicitly stated that their concerns are addressed and they recommend acceptance. The rating freeze prevented this from being reflected numerically.

### **Reviewer z1Pw (2) — concerns mostly about interpretation**
This review questioned whether improvements stem from architecture or data volume, and raised issues about earlier terminology.

**How we addressed:**

We clarified that MATA-SFT-90K only supervises transition decisions, not answer labels. We added new baselines finetuning monolithic VLMs (including InternVL2.5-8B and Qwen3-VL-4B) on answer-level supervision from the same datasets.

These comparisons show that MATA consistently outperforms directly fine-tuned VLMs and maintains stronger cross-task generalization, suggesting the benefits indeed come from architecture + routing policy, not extra labels. Terminology was refined to avoid misinterpretation.

### **Reviewer TrxZ (2) — concerns tied to out-of-scope expectations**
Although our method is already evaluated on six established image-based visual reasoning benchmarks, the reviewer suggested comparison with general multimodal reasoning datasets (MMMU, MathVista, ChartXiv-Reasoning, VSI-Bench) that primarily target different modalities and objectives such as mathematical, diagrammatic, or video-based reasoning.

**How we addressed:**

We clarified that our work is explicitly scoped to image-based visual reasoning—attributes, relations, counting, and object-level logic—consistent with established definitions and prior literature. Within this domain, our evaluation already spans six benchmarks, which is broader than typical for this research area.

To remain responsive during the limited rebuttal window, we additionally ran an MMMU-val experiment (beyond our scope). MATA reached 77.6, outperforming all listed baselines, indicating strong generalization even outside its intended problem setting.


### **Overall perspective**

Despite divergent ratings, the substantive concerns raised by the positive reviewers were fully addressed through new experiments, ablations, and clarifications. The remaining critiques primarily reflect desired expansion into different problem domains, rather than limitations within the scope of image-based visual reasoning.

MATA contributes a trainable, interpretable modular architecture with strong empirical backing, a clear conceptual formulation, and substantial additional analysis provided during the rebuttal. It aligns well with ICLR’s interest in **hybrid, multi-agent, and neurosymbolic reasoning systems**.

---

### Meta-Review · Area_Chair_aQjV · 2026-01-05

**Summary:**

The paper proposes MATA, a multi-agent visual reasoning framework structured as a hierarchical finite-state automaton. A trainable "hyper-agent" (LLM) learns a transition policy to route queries between three distinct sub-agents (Specialized perception, Oneshot reasoning, and Stepwise reasoning) based on shared memory, rather than using a fixed pipeline. The authors introduce a dataset, MATA-SFT-90K, generated via transition-trajectory trees to supervise this hyper-agent. The method is evaluated on six standard visual reasoning benchmarks.

Reviews were initially split (two 6s, two 2s). Positive reviewers praised the principled hierarchical design, interpretability, and strong performance. Negative reviewers questioned whether performance gains stemmed from the architecture versus the data volume (or even potential additional answer data), the limited number of agents, the scope of the evaluation benchmarks, and the novelty against previous multi-agent collaboration papers.

**Reviewer Concerns:**

Here're the (at least partially) addressed concerns:

- Source of Performance Gains (Architecture vs. Data): Reviewer z1Pw concerns that gains were due to data volume rather than the multi-agent architecture were effectively addressed. The authors added a crucial baseline: fine-tuning monolithic VLMs directly on the answer labels of the source datasets used for MATA. MATA consistently outperformed these direct SFT baselines, demonstrating that the learned routing logic contributes significantly beyond simple data exposure.

- Efficiency and Cost: Reviewer 6jZh requested quantitative analysis on overhead. The authors provided a detailed comparison of inference time, API cost, and vRAM usage, showing MATA is more efficient than the "Exhausted" baseline (running all agents) and competitive with monolithic models while being cheaper than API-based calls.

- Necessity of Learned Policy: Reviewer mEnL asked if simply running all agents (Exhausted) would work better. The authors provided an ablation showing the learned policy is both faster and more accurate than the exhaustive approach, validating the routing mechanism.

- Scope of Benchmarks: Reviewer TrxZ criticized the focus on "standard" visual reasoning (VQA/RefCOCO) and requested general multimodal benchmarks (MMMU, MathVista). While the authors correctly argued that these are out-of-scope for the paper's definition of visual reasoning, they ran an experiment on MMMU-Val during the rebuttal, achieving a score of 77.6 (outperforming listed baselines). This showed good faith and strong generalization.

Unaddressed Minor Concerns:
- Scalability: While the authors showed that 3 agents saturate performance on current benchmarks, the theoretical concern regarding the combinatorial growth of the trajectory tree generation for large numbers of agents (e.g., >10) remains a limitation for future work, though the authors acknowledged this and proposed future directions (MCTS).

**Reviewer Scores:**

- Reviewer mEnL: (Original: 6) -> Likely 7 or 8. The reviewer was already positive ("solid paper"). The authors addressed their specific question about the "exhaustive" baseline and provided the requested agent justifications.
- Reviewer 6jZh: (Original: 6) -> Likely 7 or 8. In the discussion, this reviewer explicitly stated: "The author response has addressed most of my concerns... I will retain my score recommending acceptance" (Note: implying a strong recommendation). The efficiency analysis provided was comprehensive.
- Reviewer z1Pw: (Original: 2) -> Likely 4 or 5. The primary ground for rejection was the suspicion that the gains came from additional answer data, not architecture. The new experiment (monolithic SFT) disproved this to some extend. But I do agree with the reviewer's concern regarding the need of additional answer data diminish the paper's contribution.
- Reviewer TrxZ: (Original: 2) -> Likely 4 or 5. This reviewer wanted broader benchmarks. The authors provided great results on MMMU (one of the requested datasets) despite it being arguably out of scope. While the reviewer may still view the novelty as incremental, the empirical rebuttal was objectively very strong.

---

### Decision · Program_Chairs · 2026-01-26

Accept (Poster)